# TORC1 is an essential regulator of nutrient-controlled proliferation and differentiation in *Leishmania*

Elmarie Myburgh [1✉], Vincent Geoghegan [2], Eliza VC Alves-Ferreira[2], Y Romina Nievas [2], Jaspreet S Grewal[2], Elaine Brown[2], Karen McLuskey [3] & Jeremy C Mottram [2]

## Abstract

*Leishmania* parasites undergo differentiation between various proliferating and non-dividing forms to adapt to changing host environments. The mechanisms that link environmental cues with the parasite's developmental changes remain elusive. Here, we report that *Leishmania* TORC1 is a key environmental sensor for parasite proliferation and differentiation in the sand fly-stage promastigotes and for replication of mammalian-stage amastigotes. We show that *Leishmania* RPTOR1, interacts with TOR1 and LST8, and identify new parasite-specific proteins that interact in this complex. We investigate TORC1 function by conditional deletion of *RPTOR1*, where under nutrient-rich conditions RPTOR1 depletion results in decreased protein synthesis and growth, G1 cell cycle arrest and premature differentiation from proliferative promastigotes to non-dividing mammalian-infective metacyclic forms. These parasites are unable to respond to nutrients to differentiate into proliferative retroleptomonads, which are required for their blood-meal induced amplification in sand flies and enhanced mammalian infectivity. We additionally show that *RPTOR1*[−/−] metacyclic promastigotes develop into amastigotes but do not proliferate in the mammalian host to cause pathology. RPTOR1-dependent TORC1 functionality represents a critical mechanism for driving parasite growth and proliferation.

**Keywords** RPTOR1; TORC1; *Leishmania*; Proliferation; Differentiation
**Subject Categories** Metabolism; Microbiology, Virology & Host Pathogen Interaction; Signal Transduction

## Introduction

*Leishmania* parasites are responsible for a group of neglected tropical diseases, termed leishmaniases. Their clinical manifestations span a broad range of severity and include self-resolving cutaneous ulcers, debilitating mucocutaneous lesions and lethal systemic disease. The disease group affects the poorest communities with an estimated 700,000–1 million new cases each year, and around 1 billion people at risk of infection (World Health Organization (WHO), 2022). The causative protozoan parasites of over 20 *Leishmania* species are transmitted between mammalian hosts by bites of infected female phlebotomine sand flies. The cycling between a mammalian host and insect vector, and movement between different niches within a single host, expose parasites to dramatic changes in environment. In the sand fly, *Leishmania* migrate through different parts of the digestive tract and develop from proliferating procyclic promastigotes to infective non-dividing metacyclic promastigotes that are pre-adapted for survival in their mammalian host (Bates, 2007; Dostálová and Volf, 2012). After transmission to the new mammalian host these highly motile flagellated promastigotes again experience changes in temperature, pH, nutrients and host-derived factors, and transform to intracellular amastigotes. Successful transmission and survival of parasites relies on their rapid adaptation to these many changes through efficient sensing of the environment and triggering of the appropriate response to drive proliferation, differentiation and/or quiescence.

Eukaryotes, including plants, yeasts, worms, flies and mammals control cell growth and differentiation in response to nutrients and environmental cues through the atypical serine/threonine kinase, TOR (target of rapamycin) (Saxton and Sabatini, 2017). TOR is highly conserved and is found in multiprotein TOR complexes (TORCs) which differ in their components, regulation and function (Sengupta et al, 2010). In mammals, a single TOR forms part of two complexes, TORC1 and TORC2, while the yeast *Saccharomyces cerevisiae* has two TOR orthologues that participate in TORCs with equivalent functions to those in mammals (Loewith and Hall, 2011). *Leishmania*, like the closely related kinetoplastid *Trypanosoma brucei*, has four TOR orthologues, TOR1 to TOR4. TOR1 and TOR2 are most likely essential in *Leishmania* promastigotes and have not been characterized, while TOR3 is required for mammalian infectivity and acidocalcisome biogenesis (Baker et al, 2021; Madeira da Silva and Beverley, 2010). *Leishmania* TOR4 remains unexplored but it was shown in *T. brucei* to form part of a complex, TORC4, which negatively regulates differentiation into quiescent stumpy form parasites that are pre-adapted for infection of their tsetse fly vector (Barquilla et al, 2012). TORC1 has been studied extensively in many eukaryotic organisms; it senses

¹York Biomedical Research Institute, Hull York Medical School, University of York, York YO10 5DD, UK. ²York Biomedical Research Institute, Department of Biology, University of York, York YO10 5DD, UK. ³Wellcome Centre for Integrative Parasitology, Institute of Infection, Immunity and Inflammation, College of Medical Veterinary and Life Sciences, University of Glasgow, Glasgow G12 8TA, UK. ✉E-mail: elmarie.myburgh@york.ac.uk

intracellular nutrient and energy levels to regulate anabolic and catabolic processes. In addition to TOR and its constitutive binding partner, mLST8 (mammalian lethal with Sec13 protein 8), TORC1 contains a complex-specific 150-kDa protein named RPTOR (regulatory-associated protein of mTOR, Kog1 in *S. cerevisiae*). This adapter protein most likely arose with TOR in the last eukaryotic common ancestor (LECA) and has been conserved with TOR during further evolution of eukaryotes (Tatebe and Shiozaki, 2017; van Dam et al, 2011). Due to its role in recruitment of substrates for phosphorylation by TOR, RPTOR is essential for TORC1 functions (Nojima et al, 2003; Schalm et al, 2003).

In this study, we defined the TORC1 complex in *Leishmania* and investigated its function through the DiCre-based conditional gene deletion (Andenmatten et al, 2013; Duncan et al, 2016) of *RPTOR* (named *RPTOR1)*. Our proteomic analyses demonstrate that *Leishmania* RPTOR1 interacts with TOR1 and LST8, confirming it as a component of *Leishmania* TORC1. Deletion of *RPTOR1* reveals that TORC1 is required for cell proliferation and long-term survival of promastigotes in vitro and is critical to establishing infections in vivo. Notably, *RPTOR1* deletion caused differentiation of procyclic promastigotes to non-dividing metacyclic promastigotes and prevented differentiation of metacyclic promastigotes to dividing retroleptomonads. In addition, complementation with mutant versions of RPTOR1 revealed that a potentially catalytic histidine cysteine dyad in its caspase-like domain is not required for RPTOR1 function showing that it is a pseudopeptidase.

## Results

### *Leishmania* RPTOR1 interacts with TOR1

To define the TORC1 complex in *Leishmania* we generated *L. mexicana* lines expressing N-terminally Twin-Strep-tagged RPTOR1 (LmxM.25.0610), TOR1 (LmxM.36.6320) or the control bait LmxM.29.3580 using a CRISPR-Cas9 mediated endogenous tagging approach (Beneke et al, 2017) (Fig. EV1A). RPTOR1 or TOR1 were enriched from parasite lysates with MagStrep XT, allowing specific elution with biotin and quantitative analysis of complexes by mass spectrometry (Fig. 1A,B, Fig. EV1B and Dataset EV1). A RPTOR1-TOR1 interaction was detected in both pull-downs and LST8 co-enriched with both RPTOR1 and TOR1 indicating that *Leishmania* RPTOR1 associates with two proteins that are key components of TORC1 in other eukaryotes. The TOR1 pull-down also detected an HSP90 chaperone (referred to as HSP83-1), the co-chaperone HIP (HSC-70 interacting protein, LmxM.08_29.0320) and two additional proteins that have been linked to TOR signalling or regulation in other organisms, RUVBL1 (LmxM.33.3500) and RUVBL2 (LmxM.33.2610), which are homologues of the ATPases Pontin (RUVBL1) and Reptin (RUVBL2). Interestingly, two proteins with no known function (LmxM.16.1080 and LmxM.08_29.1470) were significantly enriched in both the RPTOR1 and TOR1 purifications, indicating that they might form part of the core TORC1 complex in *Leishmania*. Homology and protein domain searches with the 646 amino acid sequence of LmxM.16.1080 revealed homologous sequences in *Leishmania* and *Leptomonas spp*; however, no protein domains were predicted. LmxM.08_29.1470 contains an ankyrin repeat domain suggesting a role in protein-protein interaction, and

orthologues of this 711 amino acid protein are present in several kinetoplastids including *Trypanosoma*, *Leishmaniiae*, *Strigamonadinae* and *Bodo saltans*.

To confirm the interaction of RPTOR1 and TOR1 we co-expressed RPTOR1-HA and TOR1-Myc in *L. mexicana* (Fig. EV1C) and performed western blotting on immuno-precipitated complexes (Figs. 1C and EV1D). *L. mexicana* with untagged RPTOR1, or parasite lines singly expressing RPTOR1-HA (Fig. 1C) or Myc-TOR (Fig. EV1D) were included as controls. Our analysis showed that TOR1-Myc is immunoprecipitated with RPTOR1-HA confirming that these two proteins are part of a TORC1 complex.

We next tagged the endogenous RPTOR1 with mNeonGreen for localisation of the protein in *L. mexicana* promastigotes. Because TORC1 localises to lysosomal membranes in mammalian cells (Sancak et al, 2008) and vacuolar membranes in yeast (Kira et al, 2016), we treated live cells with FM 4-64 to label endocytic compartments and the lysosome, or a lysotracker probe (Lysotracker Red DND-99) which labels acidic compartments such as acidocalcisomes in *Leishmania* promastigotes (Mullin et al, 2001). Fluorescence microscopy of live log-stage promastigotes demonstrated low RPTOR1 expression and variable localisation in different cells (Fig. 1D). RPTOR1 localised to the lysosome in parasites but was also detected in other endomembranes, the cytosol and acidic organelles.

### RPTOR1 is essential for cell proliferation

Previous unsuccessful attempts to generate a *TOR1* null mutant in *Leishmania* promastigotes suggests that TOR1 is essential for sand fly stage parasites (Baker et al, 2021; Madeira da Silva and Beverley, 2010). To investigate whether RPTOR1 is also essential, we attempted to generate *RPTOR1* null mutants in *L. major* and *L. mexicana* promastigotes. Multiple attempts to replace *RPTOR1* in *L. major* using standard homologous recombination and in *L. mexicana* using CRISPR-Cas9 mediated homologous recombination failed to generate homozygous knockout mutants despite the generation of heterozygous mutants. These observations suggested indirectly that RPTOR1 may be essential for promastigote survival. To provide direct evidence for essentiality and explore the functional consequence of *RPTOR1* deletion we made use of a dimerizable Cre (DiCre) conditional gene deletion system (Andenmatten et al, 2013; Duncan et al, 2016) which excises *LoxP* flanked (floxed) *RPTOR1* following rapamycin treatment (Fig. EV2A). A DiCre expressing *L. major* cell line (DiCre) was modified to replace one *RPTOR1* allele with a floxed GFP-tagged *RPTOR1* expression cassette (*RPTOR*[+/flox]) and the 2nd allele with an antibiotic (hygromycin) resistance cassette (*RPTOR1*[-/flox]). Diagnostic PCR amplification confirmed the integration of the knockout and expression cassettes in several hygromycin and puromycin double resistant clones (Fig. EV2B) and two of these clones, clones 1 and 2, were selected for further analyses (Fig. EV3A). PCR showed that rapamycin treatment of these *RPTOR1*[-/flox] promastigotes for three days induced excision of the *RPTOR1* coding sequence (CDS) (Fig. 2A). Rapamycin is well known as an inhibitor of mammalian TOR, however in trypanosomatids the residues important for rapamycin binding are mutated, making them relatively insensitive to rapamycin inhibition of TOR (Barquilla et al, 2008; Madeira da Silva and Beverley, 2010). To ensure we could distinguish between *RPTOR1* deletion and other potential rapamycin-specific effects, we

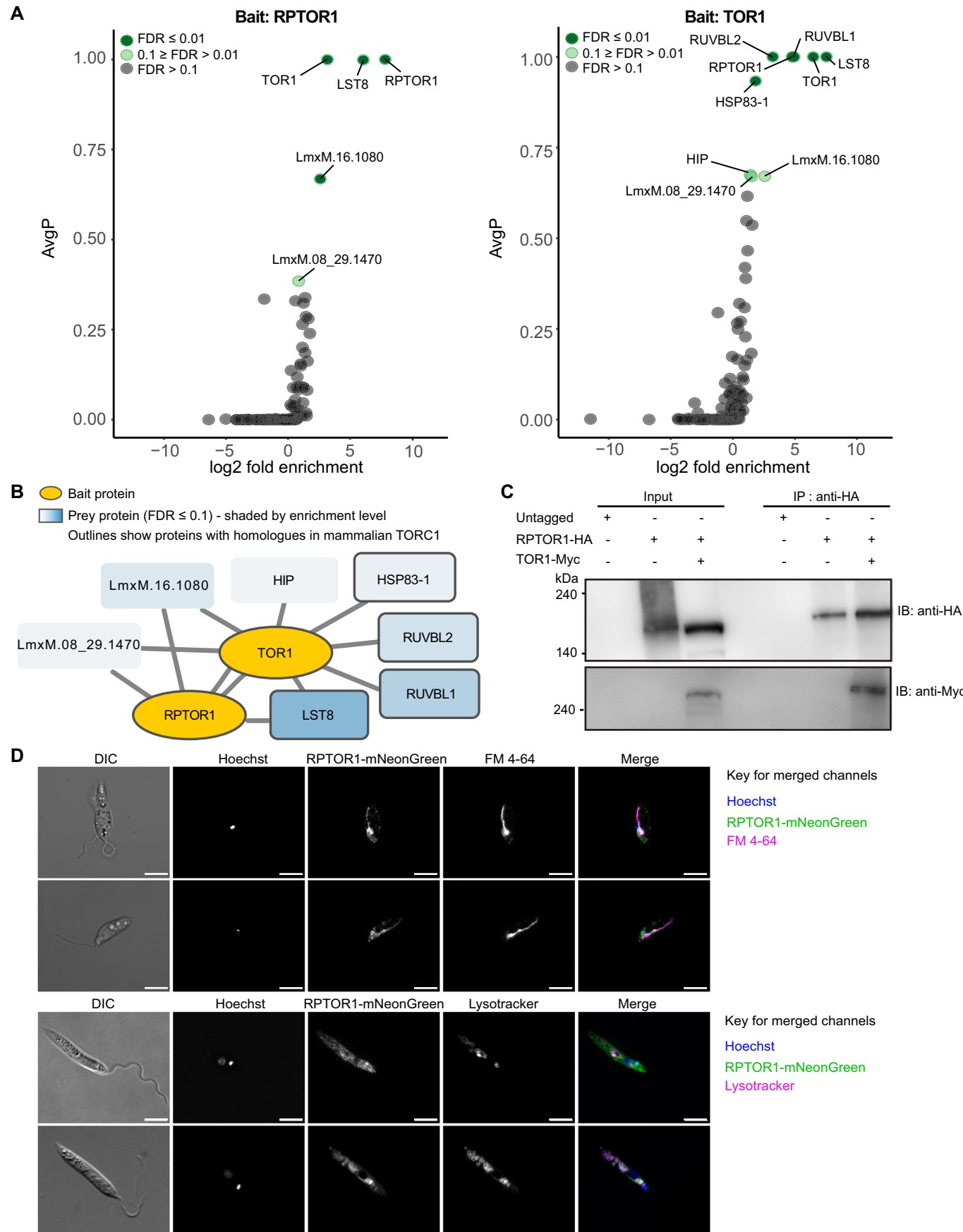

Figure 1. **Characterization of the RPTOR1 containing complex in *Leishmania*.**

(A, B) Proteins associated with RPTOR1 and TOR1 in *L. mexicana* were identified by mass spectrometry analyses of Twin-Strep-tagged RPTOR1 or TOR1 affinity purification. Volcano plots (A) show the log2 fold enrichment and SAINT scores (average probability across three biological replicates - AvgP) of identified proteins when RPTOR1 (left) or TOR1 (right) is used as bait compared to the control bait protein LmxM.29.3580. Non-significant proteins (FDR > 0.1) are indicated in grey and significant proteins are indicated in green (light green for 0.1 ≥ FDR > 0.01; dark green for FDR ≤ 0.01) based on stringent filtering criteria (see Methods) and SAINT analyses. The interaction diagram (B) shows significant interactors (FDR ≤ 0.1). Bait proteins are coloured yellow and prey proteins are shaded blue according to enrichment level, darker indicates higher fold enrichment. Proteins with outlines are homologous to members of mammalian TORC1. (C) Validation of RPTOR1-TOR1 interaction by co-immunoprecipitation. Lysates of *L. mexicana* cells expressing untagged RPTOR1 and TOR1, HA-tagged RPTOR1 and /or Myc-tagged TOR1 were incubated with anti-HA-conjugated magnetic beads. Input samples were removed before addition of beads and 20 μg of protein loaded for each sample. After 6 washes, beads were eluted in 50 μL laemmli buffer per line and half of this eluate analysed by immunoblot (IB) using anti-Myc or anti-HA antibodies. (D) Localisation of RPTOR1 using live cell fluorescence microscopy. Log-stage *L. mexicana* promastigotes expressing mNeonGreen-tagged RPTOR1 were stained with Hoechst and FM 4-64 (top two panels) or Lysotracker Red DND-99 (bottom two panels). Scale bar is 5 μm. Source data are available online for this figure.

also included the control cell line (DiCre) treated with rapamycin in all experiments. Flow cytometry of live promastigotes indicated a loss of RPTOR1-GFP protein in inducible knockout cell lines (*RPTOR1*[-/flox] clones 1 and 2) following rapamycin treatment (Fig. 2B). The GFP signal in both clones was reduced to the level detected in the DiCre control cells.

To characterise the essential role of RPTOR1 for *Leishmania* survival we started our analyses by looking at parasite proliferation in our *L. major* lines. Excision of *RPTOR1* resulted in a significant decrease in promastigote proliferation compared to uninduced (DMSO-treated) *RPTOR1*[-/flox] cells and the parental cell line, DiCre (Figs. 2C and EV3B). Cell densities in the rapamycin-treated lines were 30–40% lower on day 4 and 70–80% lower by day 7 after treatment compared to the controls. No difference in proliferation was observed for a heterozygote line (*RPTOR1*[+/−]) compared to DiCre (Fig. EV3B). To assess the effect of RPTOR1 loss on long-term parasite survival we used a clonogenic assay and found that rapamycin-induced RPTOR1 depletion caused an 80–96% reduction in clone survival (Fig. 2D). Furthermore, PCR analysis showed that all surviving clones retained a copy of *RPTOR1* (Fig. EV3C) confirming an essential role for RPTOR1 in parasite survival.

## Loss of RPTOR1 results in cell cycle arrest in G1

We next explored the reason for reduced cell proliferation and survival in the absence of RPTOR1. Flow cytometry of live cells stained with propidium iodide indicated that *RPTOR1* excision did not cause a significant amount of cell death on days 5 and 6 after induction (Figs. 2E and EV3D). However, induced *RPTOR1*[-/flox] cells arrested in the G1 phase of the cell cycle; there was a 10–20% increase of cells in G1, a 11–17% reduction in S compared to the uninduced control cells on day 5 (Fig. 2F). At later time points (day 8) both uninduced and rapamycin-induced DiCre and *RPTOR1*[-/flox] lines were arrested in G1, consistent with the predicted development of non-proliferative metacyclic promastigotes in a late stage nutrient-depleted in vitro culture (Fig. EV3E) (Mallinson and Coombs, 1989). Collectively, these results suggest that RPTOR1 plays an essential role in *L. major* promastigote cell proliferation and survival, and that its loss results in cell cycle arrest in G1. Importantly, we observed the same cell proliferation defect and cell cycle arrest in our two independently generated *RPTOR1*[-/flox] clones.

## RPTOR1 controls protein synthesis and cell size

TORC1 signalling in humans regulates protein synthesis and cell growth. We asked whether RPTOR1 depletion impacts these essential processes in *Leishmania*. Protein synthesis, measured by

incorporation of the synthetic methionine analogue L-azidohomoalanine (AHA), was reduced by 51% and 90% in *RPTOR1*[-/flox] clones 1 and 2, respectively following rapamycin-induced loss of RPTOR1 (Fig. 3A,B). To analyse cell growth, the size of promastigote parasites was measured by forward light scatter using flow cytometry on day 5 and 8 after induction. In this protocol cells from log-phase promastigote cultures were diluted in fresh nutrient-rich medium on day 3 (Fig. EV3A) so that control cells should be actively proliferating and exhibiting normal growth at day 5. By day 8 media is more depleted of nutrients and parasites would have differentiated to metacyclic promastigotes, which are smaller in size compared to the proliferating procyclic promastigotes of day 5. Forward scatter was reduced following *RPTOR1* excision (Fig. 3C,D); on day 5 the geometric means in *RPTOR1*[-/flox] lines decreased by 60–65% compared to the uninduced controls, and by 37–43% compared to the DiCre control. Two cell populations based on size were visible in the induced *RPTOR1*[-/flox] lines with 23.6–40.4% and 71.4–78.7% smaller cells measured on day 5 and 8, respectively. By day 8 the DiCre cells had also reduced in size consistent with the development of smaller metacyclic promastigotes, but these cells were still larger than the cells measured in the *RPTOR1*[-/] lines. Interestingly, the uninduced *RPTOR1*[-/flox] lines had increased forward scatter compared to the DiCre line at both day 5 and day 8 (Fig. 3D). These results indicate that RPTOR1 is important for promastigote protein synthesis and normal cell growth.

## Functional analysis of RPTOR1 caspase domain

RPTOR1 is a large protein molecule consisting of 1471 residues, composed of the characteristic domains of human RPTOR (Fig. 4A). It features a RAPTOR N-terminal conserved (RNC) region (RAPTOR N-terminal Caspase-like domain), with ~38% identity to the human RNC, an Armadillo repeat domain and C-terminal WD40 repeats, both implicated in protein-protein interactions. The RNC region is structurally similar to caspases (Ginalski et al, 2004); it contains an α-β-α sandwich fold with an apparent histidine-cysteine (His-Cys) dyad near the caspase active site. As there are no three-dimensional structures of LmRPTOR1, a structural alignment and topological comparison (Fig. 4B) of *Arabidopsis thaliana* RAPTOR1 RNC (AtRAPTOR1) with human Caspase-7 (*Hs*Caspase-7) was carried out. This showed that the secondary structural elements of AtRAPTOR1 and HsCaspase-7 are highly conserved, along with the position of the catalytic histidine and the glycine that follows it. However, in AtRAPTOR1 a serine (Ser[246]) directly aligns with the Caspase-7 active site cysteine although a cysteine (Cys[245]) does directly precede it (Fig. 4B,C; Appendix Fig. S1A) (Yang

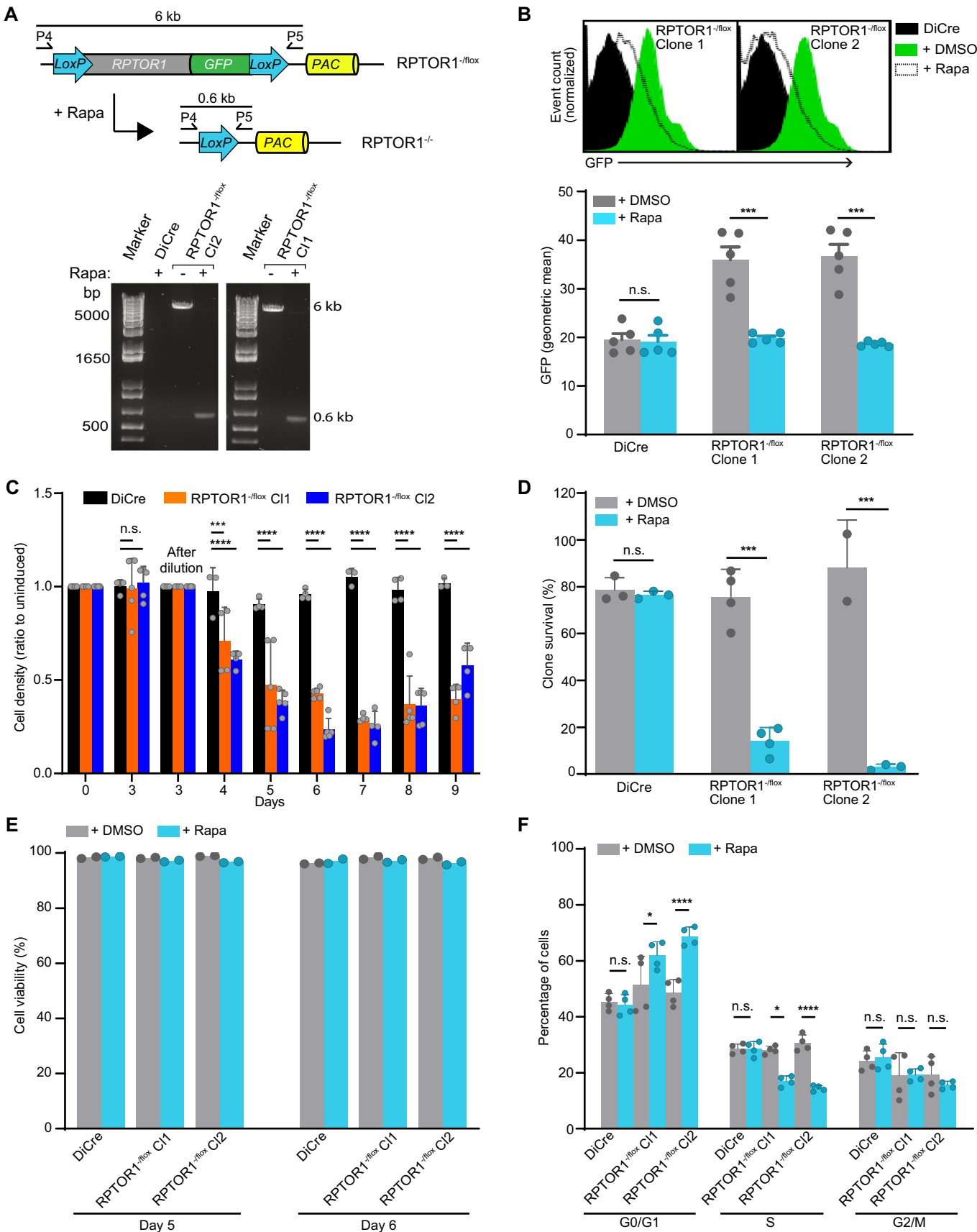

◄ **Figure 2. RPTOR1 is essential for cell proliferation.**

(A) Schematic of the *RPTOR1*-/flox locus and the expected *RPTOR1*-/- locus following rapamycin-induced recombination (top) in *L. major*. Oligonucleotide binding sites and the length of the PCR amplicons to detect *RPTOR1* excision are indicated. PCR analysis of genomic DNA from DiCre and inducible *RPTOR1*-/flox clonal lines (Cl1 and Cl2) showing recombination after three days of treatment with 100 nM Rapamycin (Rapa) added daily (bottom). (B) Analyses of GFP-tagged RPTOR1 loss by flow cytometry. Representative histograms show GFP fluorescence in live *L. major* promastigotes from untreated (+DMSO, green with solid line) and rapamycin-treated (+Rapa, white fill with dotted line) cell lines (top). Rapamycin-treated DiCre (black fill) is included in each histogram as GFP negative control. Cells were treated with DMSO or rapamycin for three days, diluted and cultured for another three days before analysis by flow cytometry. The GFP geometric mean fluorescence intensity in untreated or rapamycin-treated cells is shown (bottom). Data represent mean ± SEM from three individual experiments; each dot denotes an experiment. ***P value ≤ 0.001 in an unpaired *t* test. n.s. not significant. (C) Cell densities of rapamycin-induced relative to uninduced cells (shown as ratio to uninduced). Log-stage promastigotes of *L. major* DiCre and *RPTOR1*-/flox lines (Cl1 and Cl2) were set up at $1 \times 10^5$ cells mL⁻¹ (day 0) and treated for three days with 0.1% DMSO or 100 nM rapamycin; cells were then counted and diluted to $1 \times 10^5$ cells mL⁻¹ (day 3) followed by culturing and daily counting for five days. Data represent mean ± SD from three or four experiments (biological replicates); each dot denotes an experiment. ***P value ≤ 0.001, ****P value ≤ 0.0001 in a one-way ANOVA with Tukey's post hoc test. n.s. not significant. (D) Clone survival (shown as a percentage of plated clones) of uninduced (+DMSO) and rapamycin-induced (+Rapa) cells. Data represent mean ± SD from three to four individual experiments with two to four technical replicates each; each dot denotes the mean of replicates in an experiment. ***P value ≤ 0.001 in a one-way ANOVA with Tukey's post hoc test. n.s. not significant. (E) Cell viability of uninduced and rapamycin-induced cells. Log-stage promastigotes were treated for 3 days, then diluted and cultured for another two or three days with DMSO or rapamycin. Cells were stained with propidium iodide and analysed by flow cytometry. Data represent mean of 2 biological replicates, each dot denotes a replicate. (F) Cell cycle analysis of uninduced and rapamycin-induced cells from *L. major* DiCre, *RPTOR1*-/flox Cl1 and *RPTOR1*-/flox Cl2 lines. Promastigotes were fixed, stained with propidium iodide and analysed by flow cytometry. Data represent mean ± SD (*n* = 4 biological replicates from 2 experiments); each dot denotes a replicate. *P value ≤ 0.05, ****P value ≤ 0.0001 in a two-way ANOVA with Tukey's post hoc test. n.s. not significant. Source data are available online for this figure.

et al, 2017). Primary sequence alignment of RPTOR1 from *A. thaliana*, *H. sapiens* and *L. major* indicate that these regions are also conserved in the human and *L. major* proteins (Appendix Fig S1B) suggesting that the cysteine of the His-Cys dyad is also shifted in these other RPTOR1 proteins, with an asparagine residue (Asn²⁷⁰) occupying the position of the catalytic cysteine in *Lm*RPTOR1. Structural superposition of the Alphafold model of *L. infantum* RPTOR1 (LINF_250011400), AtRAPTOR1 and HsCaspase-7 using UCSF Chimera confirmed our results above (Fig. EV4A,B).

To assess whether *Leishmania* RPTOR1 requires the His-Cys dyad in the caspase-like RNC for its function we generated complementation lines in *L. major* *RPTOR1*-/flox clone 2 by integrating untagged and HA-tagged wildtype and cysteine (Cys²⁶⁹) mutant *RPTOR1* in the ribosomal locus (Fig. EV4C). The cysteine at position 269 in the predicted His-Cys dyad was mutated to an alanine (C269A). This resulted in the generation of Cl2::pRib-RPTOR1, Cl2::pRib-RPTOR1-HA, Cl2::pRib-RPTOR1^C269A and Cl2::pRib-RPTOR1^C269A-HA. Western blot analyses were performed with either anti-HA antibodies or antibodies raised against a recombinant 462 amino acid *L. major* RPTOR1 fragment. This detected expression of RPTOR1 or RPTOR1-HA following rapamycin-induced excision of floxed *RPTOR1* in the endogenous locus of the complementation lines (Fig. 4D). Expression of RPTOR1, RPTOR1-HA, RPTOR1^C269A or RPTOR1^C269A-HA restored parasite proliferation following rapamycin-induced excision of floxed *RPTOR1* (Fig. 4E). We also observed rescue of the cell size defect and G1 cell cycle arrest with RPTOR1 or RPTOR1^C269A re-expression (Fig. 4F,G). These results indicate that re-expression of RPTOR1 restores the phenotypes observed after rapamycin-induced excision of floxed RPTOR1 and provides confidence that these phenotypes are specific for RPTOR1 deletion. Furthermore, the disruption of the Cys²⁶⁹ in the caspase-like domain does not impact on complementation of RPTOR1 function suggesting that this residue is not required for promastigote proliferation and growth.

## RPTOR1 loss induces parasite differentiation into metacyclic promastigotes

*Leishmania* promastigotes undergo several biochemical and morphological changes in their sand fly hosts to eventually differentiate into mammalian-infective metacyclic promastigotes.

This differentiation process, termed metacyclogenesis, is induced in vitro by low pH, depletion of adenosine and low levels of tetrahydrobiopterin (Bates, 2008; Bates and Tetley, 1993; Sacks and Perkins, 1984; Serafim et al, 2012) but the mechanisms involved and the exact triggers for differentiation remain poorly defined. Our data showed that loss of RPTOR1 resulted in an increase in parasites with characteristics of metacyclic promastigotes: cells were non-proliferative, arrested in G1 and smaller. Manual counting of live cells also revealed that *RPTOR1*-/- parasites were highly motile. Since RPTOR1 has been shown to play a role in nutrient sensing in other organisms, we investigated whether it may be involved in nutrient sensing to coordinate continued proliferation or metacyclogenesis in *Leishmania*. We first analysed morphology by measuring cell size using scanning electron microscopy (SEM) of one of our *RPTOR1*-/flox clones (clone 2) (Fig. 5A,B). SEM images revealed that the difference in cell size observed by flow cytometry (Fig. 3C,D) is related to a decrease in body width of parasites (mean ± SD, 1.5 ± 0.3 μm in rapamycin-induced compared to 2.0 ± 0.3 μm in uninduced). In addition, body and flagellum length were increased in *RPTOR1*-/- cells with body lengths of 9.4 ± 2.8 μm for rapamycin-induced compared to 7.5 ± 1.9 μm for uninduced cells and flagellum lengths of 13.8 ± 4.2 μm for rapamycin-induced compared to 8.1 ± 3.9 μm for uninduced cells. Rapamycin treatment of DiCre control cells resulted in an increase in body width and a decrease in flagellum length.

We next assessed whether *RPTOR1* deletion triggered metacyclogenesis by measuring modification of lipophosphoglycan (LPG) and expression of metacyclic stage-specific proteins. LPG structure is modified during metacyclogenesis leading to a decrease in agglutination of parasites by peanut lectin (PNA) (Sacks et al, 1985). As a positive control we included stationary-phase DiCre cells that had been grown for seven days to indicate the percentage of metacyclic promastigotes purified from a nutrient-depleted culture for each assay. Rapamycin-induced excision of *RPTOR1* in *RPTOR1*-/flox resulted in a 14–18-fold increase in non-agglutinated (PNA⁻) parasites compared to DiCre after three days of culture in nutrient-rich medium, with 3.5% and 2.7% PNA⁻ cells in *RPTOR1*-/flox clone 1 and 2, respectively (Fig. 5C). In contrast, rapamycin-induced DiCre grown in the same nutrient-rich

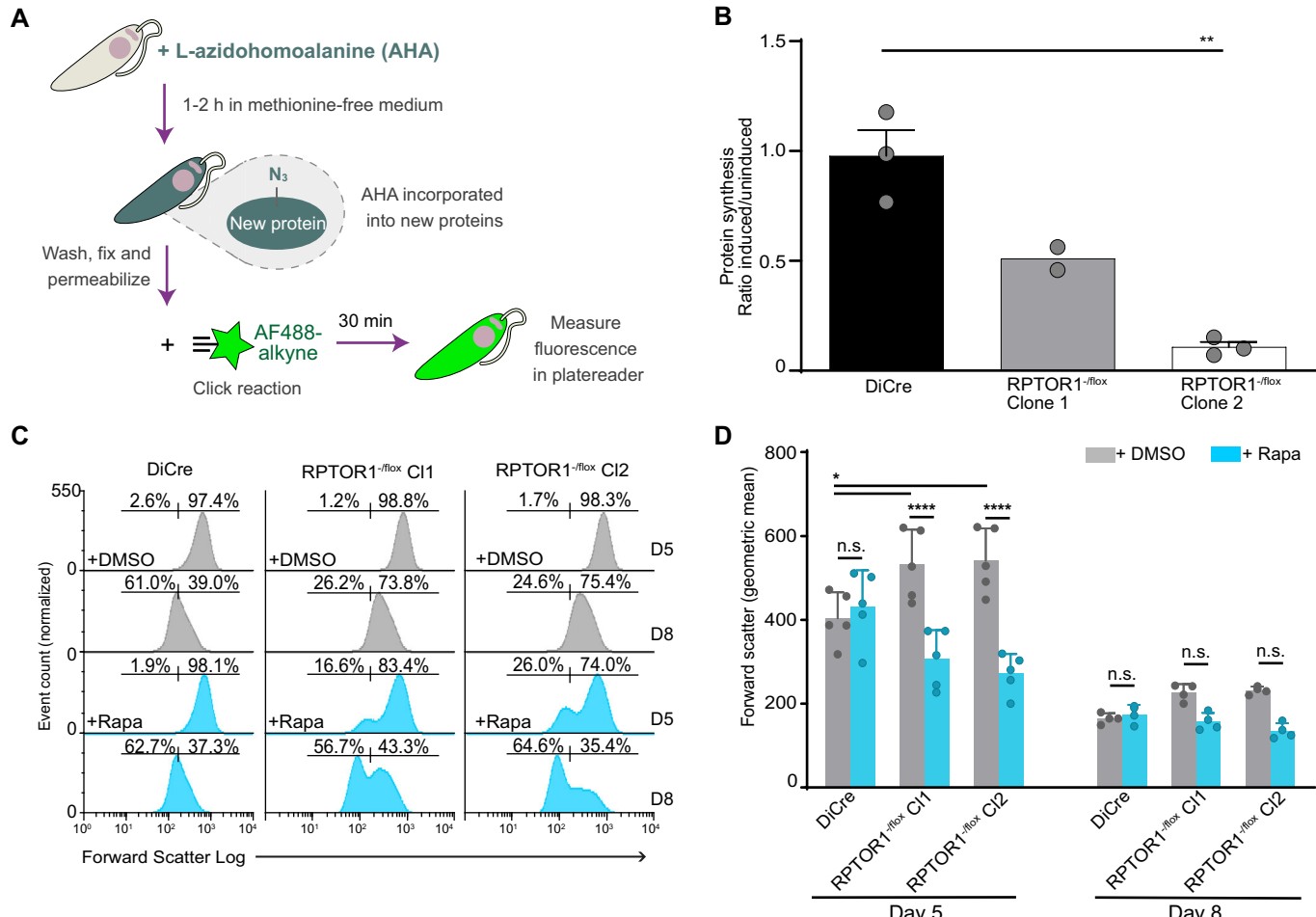

**Figure 3. RPTOR1 controls protein synthesis and cell size.**

(A) Newly synthesised proteins were measured by L-azidohomoalanine (AHA) incorporation using a Click-iT assay. AHA containing proteins were labelled with AF-488 using a click reaction and fluorescence measured in a plate reader. (B) Reduction in protein synthesis following rapamycin-induced *RPTOR1* excision in *L. major* promastigotes. Cells were harvested after six days of daily induction. The ratio of AHA incorporation in rapamycin-induced relative to uninduced (DMSO-treated) cells is shown. Data represent mean ± SEM from two or three individual experiments (biological replicates); each dot denotes an experiment. (C) Cell size was measured by flow cytometry in fixed cells after five (D5) or eight (D8) days of treatment with DMSO or rapamycin (Rapa). Histograms of forward scatter from one representative sample of each condition are shown to indicate cell size. Percentages of smaller versus larger cells are shown above the gates for each of these populations. (D) Forward scatter fluorescence intensity of indicated lines after five and eight days of treatment. Data represent geometric mean ± SD of pooled data from three or four individual experiments (biological replicates), each dot denotes an experiment. For *B to D* cells were diluted in fresh medium after the first three days of treatment. *$P$ value ≤ 0.05, **$P$ value ≤ 0.01, ****$P$ value ≤ 0.0001 in a one-way ANOVA with Tukey's post hoc test, n.s. not significant. Source data are available online for this figure.

conditions had only 0.19% PNA⁻ cells compared to 0.33% in uninduced DiCre cells. Re-expression of RPTOR1 or RPTOR1^C269A restored the phenotype back to levels in the DMSO-treated *RPTOR1^-/flox* cells. Furthermore, *RPTOR1^-/-* cells that had been continuously cultured for seven days with rapamycin had 2.6- and 4-fold more PNA⁻ cells in clone 1 and 2, respectively, compared to DiCre stationary-phase cells (also grown for seven days) (Fig. 5D). Interestingly, DMSO-treated *RPTOR1^-/flox* cells had significantly less PNA⁻ cells than DMSO-treated DiCre for both the 3 and 7-day culture conditions, with ~0.002% and 0.2–1.4% PNA⁻ for *RPTOR1^-/flox* cells in the respective culture conditions compared with 0.3% and 8.4% for DiCre cells (Fig. 5C,D). This suggests that *RPTOR1^-/flox* cells show a reduction in metacyclogenesis compared to control DiCre cells. We also measured expression of the metacyclic stage-specific protein, SHERP (small hydrophilic endoplasmic reticulum-

associated protein) (Inbar et al, 2017) using flow cytometry. Loss of RPTOR1 resulted in a significant increase in cells with high SHERP expression while *RPTOR1^-/flox* cells had lower SHERP expression compared to the DiCre control (Figs. 5E, EV5A and Appendix S2). These data show that loss of RPTOR1 gives rise to parasites with morphological and surface characteristics of metacyclic promastigotes. It also shows that the altered *RPTOR1* gene in the *RPTOR^-/flox* line results in inhibition of metacyclogenesis compared to control DiCre cells with a WT *RPTOR1* gene.

## Loss of RPTOR1 abrogates lesion development in mice

Next, we investigated the infectivity of these metacyclic promastigote-like cells by purifying PNA⁻ cells from DiCre control or *RPTOR1^-/flox* after rapamycin induction and assessing infection of

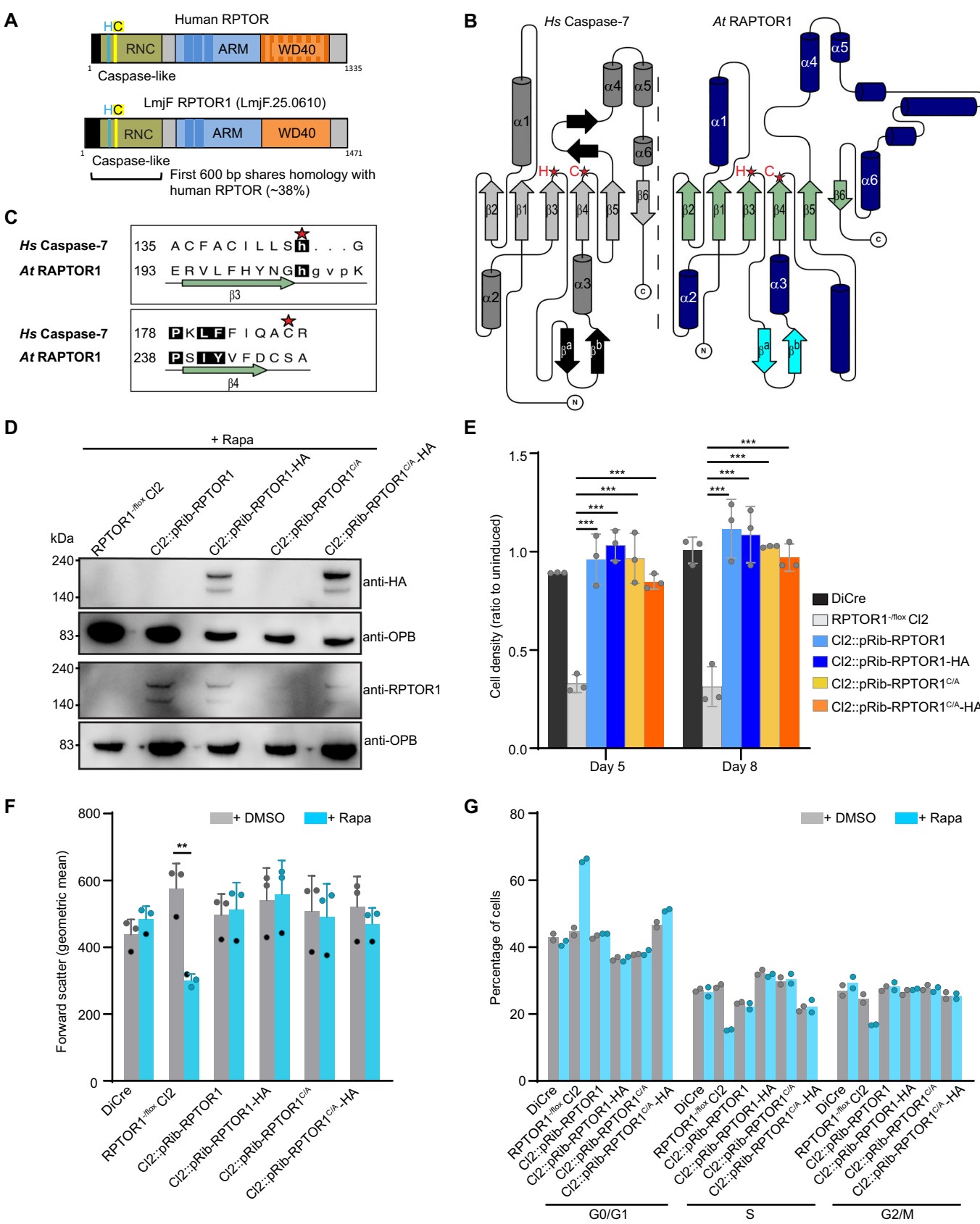

◄ **Figure 4. Functional analysis of RPTOR1 caspase domain.**

(A) Schematic representation of human RPTOR and *L. major* RPTOR1 showing the caspase-like raptor N-terminal conserved (RNC) domain, armadillo repeat (ARM) and C-terminal WD40 repeat domains. The potential caspase active site histidine (H) cysteine (C) dyad is highlighted. (B) Topology diagrams representing structural domains from human caspase-7 (PDB:1F1J) and *A. thaliana* RPTOR1 (PDB:5WBI), drawn using TOPDRAW. This shows structural similarities between the caspase-like domain of human caspase-7 (left) and *A. thaliana* RPTOR1 (right). β-strands are coloured light grey or green, and α-helices are coloured dark grey or dark blue. Conserved structural elements are numbered from the N terminus and the position of the histidine-cysteine dyad is shown by red stars. (C) Sequence alignment of the histidine-cysteine dyad containing regions from human caspase-7 and *A. thaliana* RPTOR1. β-strands are coloured green; loop regions shown as a black line and the histidine and cysteine residues of the dyad are indicated for caspase-7 by red stars. Residues where the backbones of the two structures overlay tightly are shown in upper case and those with higher deviation/no matching residue are in lower case. Numbers at the start of each line indicate the position of the first residue shown in that line in the UniProt sequences. (D) Western blot analysis of lysates from rapamycin-induced (+Rapa) *L. major* cell lines confirms the expression of untagged and HA-tagged RPTOR1 in the complementation mutants and RPTOR1's absence in the knockout mutants. Cells were induced for 3 days, diluted in fresh culture media and induced for another 3 days. (E) Cell densities of rapamycin-induced relative to uninduced *L. major* DiCre, *RPTOR1⁻/flox* Cl2 and *RPTOR1* complementation lines (shown as ratio to uninduced) five or eight days after induction. The graph shows the mean ± SD of pooled data from three individual experiments; each dot denotes an experiment. \*\*\*$P$ value ≤ 0.001 in a two-way ANOVA with Bonferroni post hoc test. (F) Forward scatter fluorescence intensity of indicated lines after five days of induction. Data represent geometric mean ± SD ($n = 3$ biological replicates) from two experiments; each dot denotes a sample, matched samples from the 2nd experiment is shown with crosses. \*$P$ value ≤ 0.05, \*\*\*\*$P$ value ≤ 0.0001 in a one-way ANOVA with Tukey's post hoc test. (G) Cell cycle analysis of uninduced and rapamycin-induced *L. major* cells. Promastigotes were fixed, stained with propidium iodide and analysed by flow cytometry. Data represent mean of replicate samples ($n = 2$ biological replicates) from two experiments; each dot denotes a replicate. Source data are available online for this figure.

thioglycollate elicited peritoneal macrophages isolated from mice. Intracellular parasite numbers at 3 h post-infection (p.i.) showed similar uptake of DiCre control and *RPTOR1⁻/⁻* PNA⁻ cells by macrophages (Fig. 6A). However, at later timepoints DiCre parasite numbers had increased (1.6-fold by day 1 and 3.3-fold by day 4 compared to 3 h) while the number of *RPTOR1⁻/⁻* remained unchanged (0.9-fold by day 1 and 0.6-fold by day 4 compared to 3 h) suggesting that *RPTOR1⁻/⁻* parasites do not develop into proliferating amastigotes in peritoneal macrophages (Figs. 6A and EV5B). This could occur due to a defect in differentiation of metacyclic promastigotes to amastigotes (amastigogenesis) or normal amastigogenesis but a defect in proliferation of amastigotes. To test this, we first harvested parasites from macrophages at day 5 p.i. and assessed their morphology through measurement of parasite length and width (aspect ratio). Parasites were visualized by staining with anti-*Leishmania* oligopeptidase B (OPB) (Munday et al, 2011), which localises to the cytosol and the DNA dye DAPI. Both DiCre and *RPTOR1⁻/⁻* exhibited similar rounded amastigote morphology (Fig. 6B,C) at day 5 p.i. compared to their elongated forms as metacyclic promastigotes (0 h p.i.). We next assessed amastigote proliferation by using a Click-iT EdU assay. Infected macrophages were treated with EdU from day 2 to day 5 p.i. followed by a click reaction, to fluorescently label EdU, and staining with DAPI. EdU incorporation into parasite DNA was then assessed by fluorescence microscopy (Fig. 6D,E). We observed variable background fluorescence between host macrophages and therefore quantified each parasite's EdU signal relative to the background in its host cell. While 58% of the DiCre amastigotes imaged were EdU+ we detected only a few (3%) of EdU+ *RPTOR⁻/⁻* amastigotes. These data confirm that loss of RPTOR1 allows differentiation into amastigotes but results in an inability of amastigotes to proliferate. To confirm this proliferation defect in vivo we infected BALB/c mice in the ears with *L. major* PNA⁻ DiCre or PNA⁻ *RPTOR1⁻/⁻* parasites and monitored lesion development. Loss of RPTOR1 completely abrogated lesion development in mice, unlike the DiCre control, showing that *RPTOR1⁻/⁻* parasites cannot establish an infection or replicate as amastigotes in vivo (Fig. 6F). In addition, we assessed parasite load in the ears of these mice at 6 wks p.i. using qPCR (Fig. 6G). *RPTOR1⁻/⁻* parasite loads were significantly lower than DiCre in ears of infected mice. Compared to a positive control (an ear spiked with $1 \times 10^5$ DiCre parasites) DiCre-infected ears contained 20-900x more parasite DNA while *RPTOR1⁻/⁻*-infected ears contained on average less parasite DNA than the positive control (9 ears contained 0.6x or less parasites while 1 ear contained 2.7x more parasites) but more compared to a naïve ear. These data show that parasite DNA is still detected in the ears of *RPTOR1⁻/⁻*-infected mice at 6 wks post-infection but that parasites either decreased (9/10 mice) or increased only slightly (1/10 mice) compared to the original infection dose ($1 \times 10^5$ parasites/ear). We also performed end-point PCR on gDNA extracted from infected ears and found that escape mutants (with un-excised *RPTOR1*) could provide an explanation for the parasite DNA detected in ears of all ten mice infected with *RPTOR1⁻/⁻* parasites (Fig. EV5C).

## RPTOR1 is essential for differentiation of *Leishmania* metacyclic promastigotes to retroleptomonads

Non-dividing *Leishmania* metacyclic promastigotes have been reported to have the ability to differentiate into replicative retroleptomonads in response to a blood meal in the sand fly vector or addition of nutrients in culture (Mallinson and Coombs, 1989; Serafim et al, 2018). We hypothesized that if RPTOR1 is involved in nutrient sensing to promote cell proliferation then it may also influence this process. Our results showed that while PNA⁻ metacyclic DiCre proliferated in nutrient-rich medium, PNA⁻ *RPTOR1⁻/⁻* parasites remained non-proliferative (Fig. 6H). These data suggest that RPTOR1 is not only essential for proliferation of *L. major* leptomonad promastigotes and amastigotes but also for the nutrient-induced differentiation of metacyclic promastigotes to retroleptomonads, a critical process for maintaining infectivity of the insect vector and successful *Leishmania* transmission.

## TORC1 is downstream of the purine sensing mechanism

*Leishmania* cannot synthesise purine nucleotides *de novo* and are dependent on this nutrient's availability in their host for growth (Carter et al, 2010; Marr et al, 1978). Purine starvation, for example through the withdrawal of purines in vitro from culture medium, results in parasite growth arrest in G1/G0 of the cell cycle and metabolic changes to a quiescent-like state (Carter et al, 2010; Martin et al, 2014). Conversely, purine supplementation can

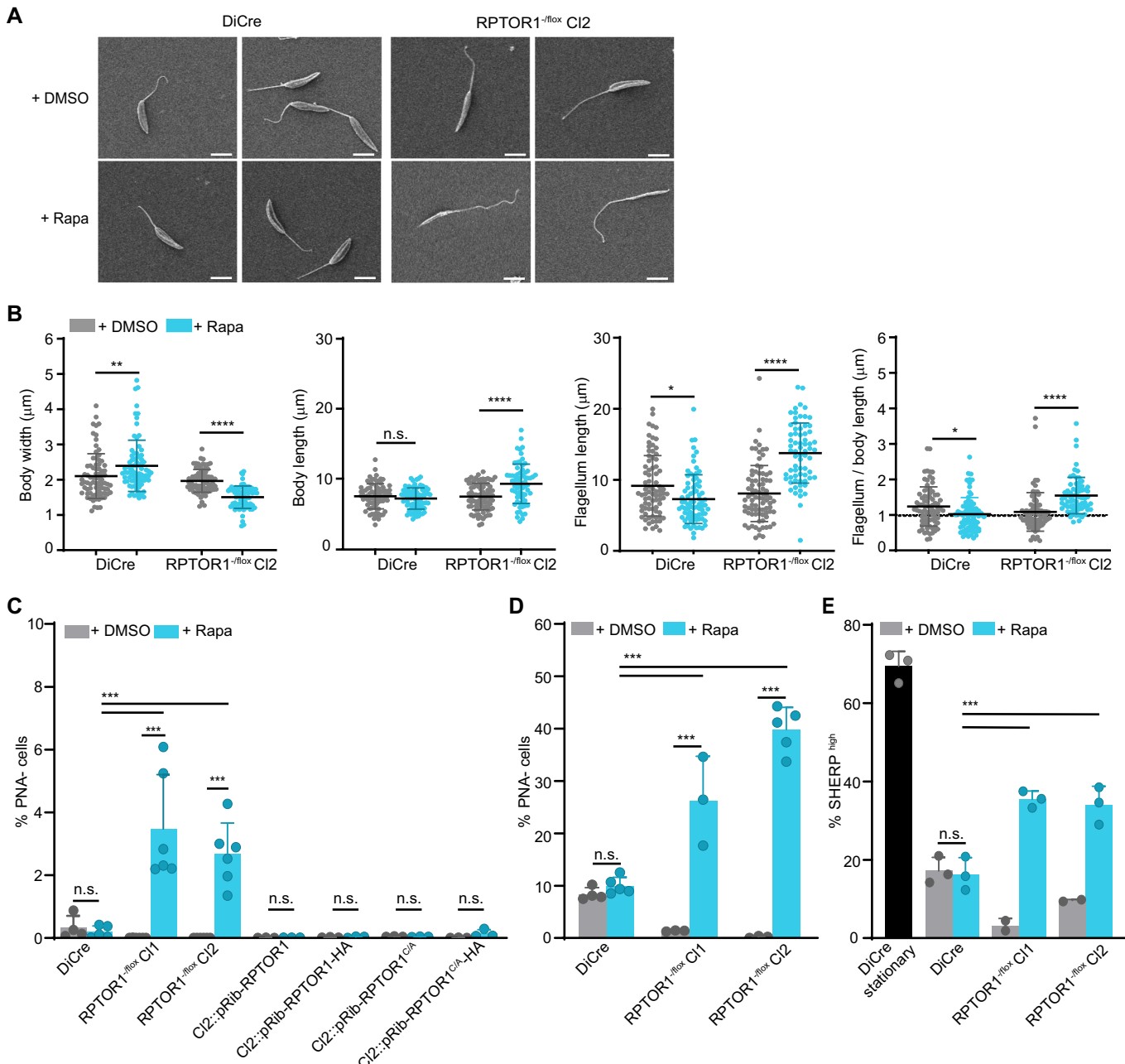

**Figure 5. RPTOR1 loss induces metacyclogenesis but is detrimental for murine infection.**

(A) Representative scanning electron microscopy images of *L. major* DiCre and *RPTOR1*⁻/flox Cl2 promastigotes after five days of treatment with DMSO or rapamycin (Rapa). Scale bar = 5 μm. (B) Body width, body length and flagellum length were measured in at least 69 cells. Means ± SD are shown, and each dot represents the measurement in a single cell. (C) Percentage of metacyclic promastigotes (PNA⁻cells) in cultures treated daily for six days with DMSO or 100 nM rapamycin (+Rapa). For A to C cells were diluted in fresh medium after the first three days of treatment. (D) Percentage of metacyclic promastigotes in cells cultured continuously for seven days with daily addition of DMSO or rapamycin (+Rapa) and no dilution. The graphs in C and D show the mean ± SD of pooled data from three to five individual experiments; each dot denotes an experiment (n = 3–5 biological replicates). (E) *RPTOR1* excision increases the percentage of cells expressing high levels of SHERP. Expression was analysed by flow cytometry in fixed cells stained with anti-SHERP and Alexa Fluor 647-conjugated secondary antibodies. Graph shows mean ± SD from pooled data of two experiments; each dot denotes a sample (n = 2–3 biological replicates). Data information: For B to E, *P value ≤ 0.05, **P value ≤ 0.01, ***P value ≤ 0.001, ****P value ≤ 0.0001 in a one-way ANOVA with Tukey's post hoc test. n.s. not significant. Source data are available online for this figure.

reverse this growth arrest. Purine sensing has also been linked to metacyclogenesis with adenosine supplementation recovering proliferation of metacyclic promastigotes in culture and inhibiting metacyclogenesis in the sand fly host (Serafim et al, 2012). We

predicted that TORC1 would function downstream of the parasite purine sensing mechanism and may be a key complex to facilitate nutrient sensing, including for purines. To investigate this, we assessed how RPTOR1 deletion influenced parasite growth

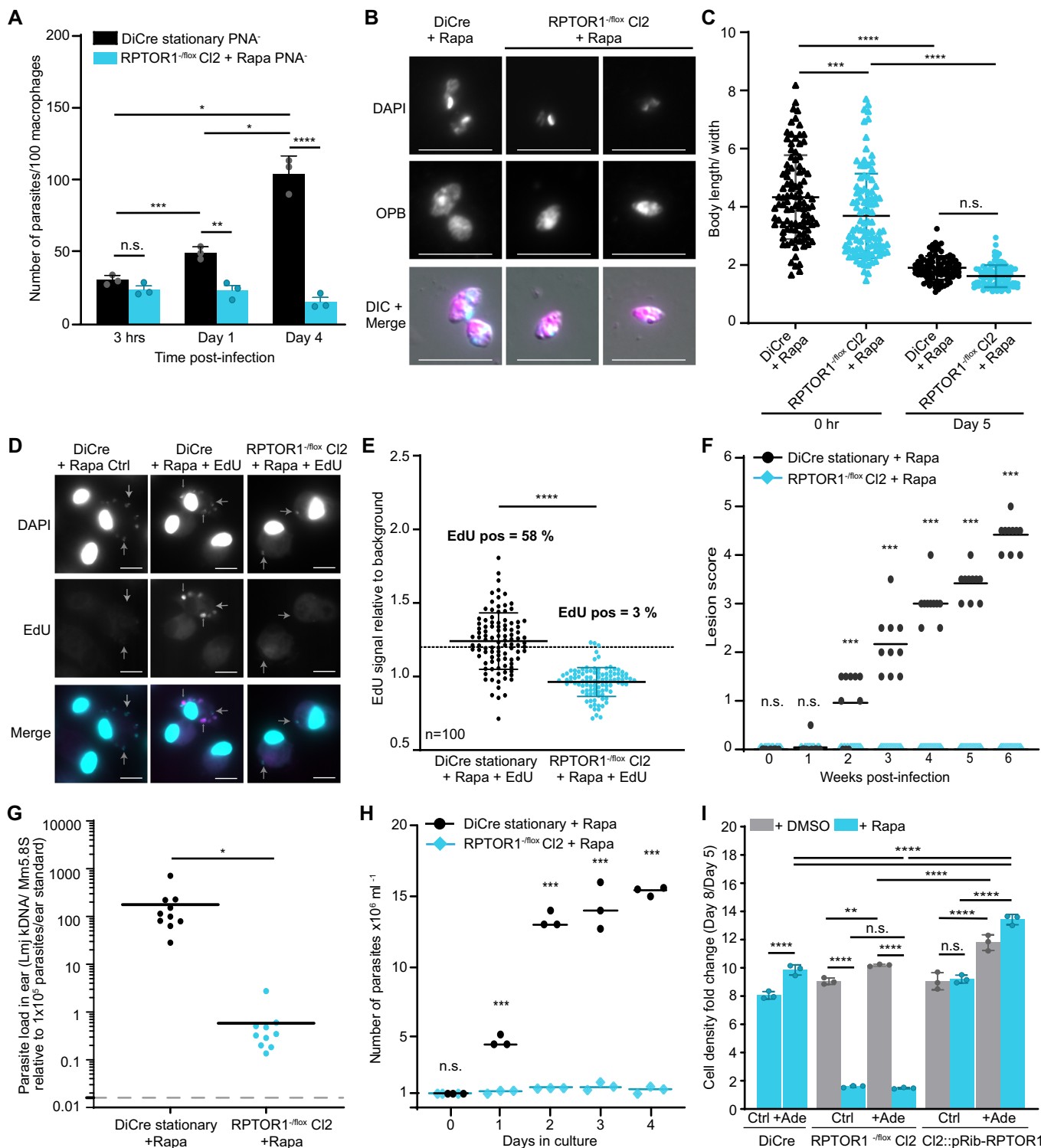

following adenine supplementation. The addition of adenine increased the proliferation of rapamycin-treated *L.major* DiCre and uninduced (DMSO-treated) *RPTOR1*-/flox parasites (Fig. 6I). On the other hand, *RPTOR1*-/- parasites showed a low level of proliferation that did not increase in response to adenine supplementation. We also included our wildtype RPTOR1 re-expression line (Cl2::pRib-RPTOR1) to assess whether higher levels

of RPTOR1 in parasites might increase responsiveness to adenine supplementation. In this case, adenine supplementation increased proliferation even more compared to the DiCre and *RPTOR1*-/- parasites. This suggests that the level of RPTOR1, and thus TORC1 activity in cells may fine-tune the responsiveness to nutrients with higher levels of RPTOR1 resulting in more proliferation of cells when nutrients are available.

◀ **Figure 6. RPTOR1$^{-/-}$ parasites can differentiate into amastigotes but are unable to proliferate in macrophages or mice.**

(A) Macrophage infectivity of PNA$^-$ *L. major* promastigotes. DiCre cells were cultured for 7 days in the presence of rapamycin to reach stationary phase. *RPTOR1$^{-/flox}$* cells were induced with rapamycin for three days, diluted in fresh medium and cultured for three more days with daily addition of rapamycin. Metacyclic promastigotes (PNA$^-$) were then purified from cultures by PNA agglutination and added to thioglycollate-elicited peritoneal macrophages at a 3:1 ratio (parasites:macrophages). Cells were analysed at 3 h, day 1 and day 4 after infection using microscopy. Graph shows values and mean ± SD of triplicate wells from one experiment. A repeat experiment using an infection ratio of 6:1 is shown in Fig. EV5B. *$P$ value ≤ 0.05, **$P$ value ≤ 0.01, ***$P$ value ≤ 0.001, ****$P$ value ≤ 0.0001 in a two-way ANOVA with Bonferroni post hoc test for analysis between groups and Tukey's post hoc test for analysis between time-points. n.s. not significant. (B) Amastigotes isolated from infected peritoneal macrophages. Metacyclic promastigotes were prepared after induction as described in (A) and used to infect peritoneal macrophages at a 6:1 ratio (parasites:macrophages). Parasites were isolated from macrophages on day 5 after infection, stained with DAPI and anti-OPB antibody, and imaged by fluorescence microscopy. Representative images of amastigotes are shown. Scale bar is 10 µm. (C) Aspect ratio (body length relative to body width) of purified metacyclic promastigotes before infection (0 h) and amastigotes isolated at day 5 after infection. Metacyclic promastigotes were prepared after induction as described in (A) and used to infect peritoneal macrophages at a 6:1 ratio (parasites:macrophages). Individual points for each parasite ($n = 100$ parasites per group) and means ± SD are shown. ***$P$ value ≤ 0.001, ****$P$ value ≤ 0.0001 in a one-way ANOVA with Tukey's post hoc test. n.s. not significant. (D, E) EdU incorporation in amastigotes within thioglycollate-elicited peritoneal macrophages at day 5 after infection. Metacyclic promastigotes were prepared after induction as described in (A) and used to infect peritoneal macrophages at a 6:1 ratio (parasites:macrophages). EdU (100 µM) or DMSO was added to infected macrophages on day 2 after infection. Cells were fixed, subjected to a click reaction using an AF647 azide as described in materials and methods, stained with DAPI and imaged by fluorescence microscopy. (D) Large arrowheads show examples of parasites that are negative for EdU while small arrowheads show parasites that are EdU positive. Scale bar is 10 µm. (E) The EdU signal in each amastigote relative to the background in its host macrophage is shown for individual parasites ($n = 100$ parasites per group); the mean ± SD is shown with a line. Cells with an EdU signal above 20% of background (shown by the dotted line) are assigned as EdU positive. ****$P$ value ≤ 0.0001 in an unpaired $t$ test. (F) Lesion development (depicted by lesion score (Schuster et al, 2014)) in BALB/c mice injected intradermally in the ear with $1 \times 10^5$ *L. major* metacyclic promastigotes ($n = 10$ mice per group combined from two experiments, individual points from each mouse and means as lines are shown). Metacyclic promastigotes were prepared after induction as described in (A) and injected into ears in PBS. ***$P$ value ≤ 0.001 in a two-way ANOVA with Bonferroni post hoc test. (G) Parasite load in ears of infected mice determined by qPCR. Individual points show relative values for each mouse and the lines show means ($n = 10$ mice per group combined from two experiments shown in (F)). The grey dotted line shows the mean background signal in two naive ears. *$P$ value ≤ 0.05 in an unpaired $t$ test. (H) *L. major* RPTOR1$^{-/-}$ metacyclic promastigotes are unable to differentiate into proliferating retroleptomonads in nutrient-rich medium. Purified metacyclic promastigotes were cultured in nutrient-rich medium. Data represent one of two similar experiments, individual points from triplicate cultures and means as lines are shown. ***$P$ value ≤ 0.001 in a two-way ANOVA with Bonferroni post hoc test. (I) Cell proliferation of rapamycin-induced cells from *L. major* DiCre, *RPTOR1$^{-/flox}$* Cl2 and *RPTOR1* complementation lines grown in Grace's medium with the addition or not of 500 µM adenine. After rapamycin treatment for five days cells were diluted (day 5) and cultured for a further three days (day 8) in the presence or not of adenine. The proliferation fold change (density at day 8 compared to day 5) is reported. Graph shows individual points and mean ± SD ($n = 3$ biological replicates) from one experiment. ****$P$ value ≤ 0.0001 in a one-way ANOVA with Tukey's post hoc test. n.s. not significant. Source data are available online for this figure.

## Discussion

The TORC1 signalling pathway in *Leishmania* is largely unexplored due to its critical role in cell growth and proliferation, and the technical challenges of functionally characterising essential genes in this organism (Jones et al, 2018; Madeira da Silva and Beverley, 2010). In this study, we confirm TORC1 essentiality through conditional gene deletion of the TOR1 binding partner, RPTOR1. Our results show that TORC1 is critical for promastigote growth and that loss of RPTOR1 triggers metacyclogenesis despite the presence of nutrients. Furthermore, it is also required for the proliferation of amastigotes in vivo and the differentiation of metacyclic promastigotes to proliferative retroleptomonads.

*Leishmania* TORC1 composition, function and regulation is unknown. Bioinformatic analysis by us and others (Lypaczewski et al, 2021; Madeira da Silva and Beverley, 2010; Tatebe and Shiozaki, 2017) identified *Leishmania* homologues for the key TOR1 complex members TOR, RPTOR1 and LST8 (also found in TORC2) but not for DEPTOR and PRAS40, which associate with these TORC1 members in vertebrates (Loewith et al, 2002; Saxton and Sabatini, 2017). Our proteomic and interaction analyses of the RPTOR1 and TOR1-containing complexes showed that *Leishmania* RPTOR1 associates with TOR1 and LST8 to form TORC1. Other TOR1 interactors included the AAA+ family ATPases RUVBL1 and RUVBL2, the molecular chaperone HSP83_1 (an HSP90 homologue) and the co-chaperone HIP, which associates with intermediate HSP90 and HSP70 complexes (Prapapanich et al, 1996). In humans RUVBL1 and RUVBL2 form part of multiprotein complexes that are involved in multiple regulatory processes (Huber et al, 2008; Mao and Houry, 2017; Nano and Houry, 2013) and loss of either RUVBL protein impairs cell growth and proliferation (Huber et al, 2008). The

*Leishmania* proteins contain the characteristic DNA-binding and ATPase motifs (Ahmad et al, 2013) but their importance for parasite growth and proliferation have not been explored. It was not surprising to find the chaperone HSP90 (named HSP83_1 in *Leishmania*) associated with TOR1—many HSP90 clients are involved in signalling, and TOR and RPTOR1 have been identified as HSP90 interactors by others (Echeverría et al, 2011). In *Leishmania*, HSP83_1 is crucial for proliferation of both promastigotes and amastigotes and its inhibition causes growth arrest and promastigote to amastigote differentiation (Hombach et al, 2013; Morales et al, 2010; Wiesgigl and Clos, 2001). RPTOR1 localised to multiple subcellular compartments, predominantly the lysosome but was also found in endomembranes, the cytosol and acidic organelles in some cells. *L. mexicana* TOR1 was reported to be localised to endomembranes and the cytosol (Baker et al, 2021), which correlates with RPTOR1 localisation supporting a RPTOR1-TOR1 interaction. Studies in other organisms showed dynamic relocation of TORC1 components depending on the nutrient availability (Kira et al, 2016; Manifava et al, 2016). This suggests that more in-depth colocalisation studies would be useful to explore *Leishmania* RPTOR1 localisation and its interaction with TOR1 in different nutrient-rich or -depleted conditions.

In Opisthokonta, TOR signalling is assessed through phosphorylation of its substrates p70 S6 kinase and 4E-BP1 (restricted to Euteleostomi), which promote protein translation through ribosome biogenesis and translational initiation of capped mRNA, respectively (Brown et al, 1995; Hara et al, 1997). Orthologues of either of these proteins have not been identified in trypanosomatids nor have other TOR1 substrates been defined. This prevented us from directly assessing TORC1 signalling through phosphorylation of its targets, and we instead measured cellular activities associated with TORC1 signalling following rapamycin-induced *RPTOR1* excision.

Our data showed that RPTOR1/TORC1 is an essential positive regulator of cell proliferation and growth of the insect-stage promastigotes. Loss of RPTOR1 induced proliferation arrest in the G1 phase of the cell cycle, inhibited protein synthesis and induced physiological changes characteristic of their differentiation into cell cycle arrested metacyclic promastigotes (Fig. 7). RPTOR1 loss in other systems such as yeast (*S. cerevisiae*) and a range of different mammalian cell types results in a G1 cell cycle arrest and either inhibits cell cycle progression in proliferating cells or prevents cell cycle entry from quiescence (Dowling et al, 2010; Rodgers et al, 2014; Yang et al, 2013). We observed these effects in log-stage promastigotes, which differentiated to non-dividing metacyclic promastigotes and were unable to re-enter cell cycle to differentiate into retroleptomonads. Importantly, metacyclogenesis is not an automatic consequence of reduced proliferation. Serafim et al showed that loss of adenosine signalling inhibits proliferation and triggers metacyclogenesis in *Leishmania* (Serafim et al, 2012). However, inhibiting the purine salvage pathway reduces proliferation but does not trigger metacyclogenesis. Our work

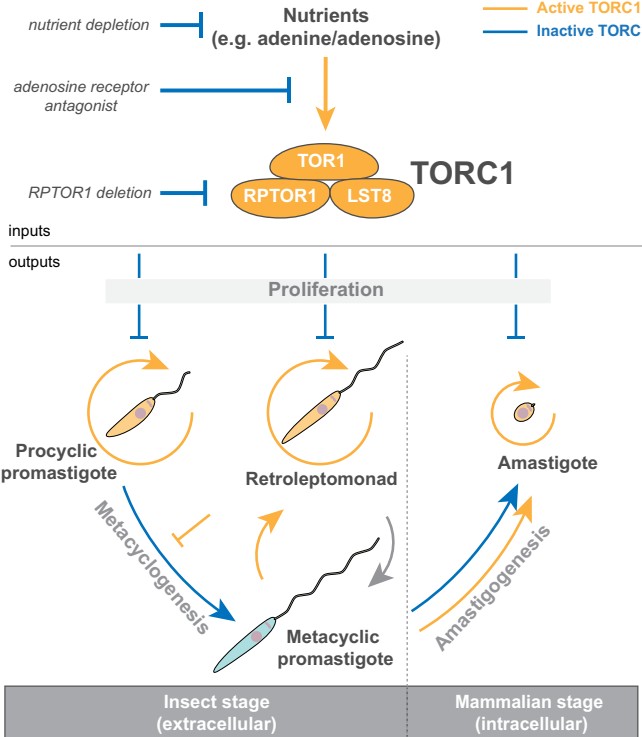

shows that deletion of *RPTOR1* produces metacyclic parasites in nutrient-rich medium, suggesting that purine or other nutrient sensing is upstream and provides a key activating signal to *Leishmania* TORC1, preventing metacyclogenesis under normal conditions. A reduction in TORC1 activity either through inducible deletion of *RPTOR1* or depletion of purines (Serafim et al, 2012), inhibits proliferation and triggers the metacyclic differentiation program (Fig. 7). It is also possible that *RPTOR1* deletion causes irreversible differentiation of parasites to metacyclic promastigotes that can no longer proliferate in response to nutrient availability through other pathways.

RPTOR1 loss also prevented the expansion of amastigotes in macrophages and in vivo in mice. Metacyclic promastigotes were phagocytosed by macrophages and differentiated to amastigotes but could not proliferate. This provides evidence that TORC1 is important for the mammalian stage of the parasite and could potentially be targeted to inhibit growth and proliferation in their human host. Further support for this comes from a recent study that investigated the Rag GTPases in the visceralizing species *Leishmania donovani* (Lypaczewski et al, 2021). The Rag GTPases (RagA/C or RagB/D heterodimers) act upstream of TORC1 to sense amino acids, and the RagA/C complex is present in *Leishmania spp*. RagA was essential for promastigote growth while RagC was not essential for promastigotes but was required for parasite survival in mice. Unfortunately, little else is currently known about the TOR pathway in *Leishmania* but the bioinformatic analyses suggest distinct differences in sensing and signalling components that could be explored for selective chemotherapy.

Initial structure prediction analysis indicated conservation of a cysteine-histidine (Cys-His) dyad in RPTOR1's N-terminal caspase-like domain suggesting that it may have peptidase activity (Ginalski et al, 2004). However, structural data of the *Arabidopsis thaliana* protein (AtRAPTOR1) provided evidence that the caspase cysteine is replaced by a serine while the adjacent cysteine faces into a hydrophobic core (Yang et al, 2017). Our primary sequence alignment indicates that the caspase cysteine is replaced by an asparagine (*Lm*, Asn[270]) in *Leishmania* (Appendix S1B) and using complementation we show that the adjacent cysteine (Cys[269]) is not required to complement the LmjRPTOR1 loss of fitness phenotype. RPTOR1 appears to be a pseudopeptidase, but the structural data for human RPTOR1 show that the caspase-like domain is optimally positioned within the TORC1 active site cleft suggesting that it may be important for substrate recognition and recruitment (Aylett et al, 2016).

In summary, our data identify TORC1 as crucial for *Leishmania* growth and proliferation in response to nutrients, for both the insect-stage promastigotes and the mammalian-stage amastigotes. It also links the nutrient-sensing TORC1 pathway to differentiation to reveal TORC1 as a key mechanism by which *Leishmania* parasites respond to nutrient availability to inhibit their differentiation from proliferating promastigotes to infective metacyclic promastigotes. This study sets a framework for further dissection of the TORC1 signalling cascade and its regulators.

**Figure 7. A proposed model of *Leishmania* proliferation or differentiation in response to nutrients and active TORC1.**

RPTOR1-dependent TORC1 is essential for the proliferation of *Leishmania* procyclic promastigotes, retroleptomonads and amastigotes. Availability of nutrients activates TORC1 to promote parasite proliferation while metacyclogenesis is inhibited (shown in orange lines). This can be enhanced through the supplementation of media with purines such as adenine or adenosine (Serafim et al, 2012). Conversely, when TORC1 is inactivated through either the depletion of nutrients, blocking of adenosine signalling (Serafim et al, 2012) or the deletion of RPTOR1, then parasite proliferation is inhibited and metacyclogenesis occurs (blue lines). This results in the differentiation of procyclic promastigotes to metacyclic promastigotes that are unable to differentiate into proliferating retroleptomonads (Serafim et al, 2018). These metacyclic promastigotes can also differentiate into amastigotes (amastigogenesis) but are unable to proliferate in the murine host.

## Methods

### Reagents and tools

See Table 1.

**Table 1. Reagents and tools.**

| Reagent/Resource | Reference or Source | Identifier or Catalog Number |
|---|---|---|
| **Experimental models** | | |
| BALB/c (*M. musculus*) | Charles River Laboratories | BALB/cAnNCrl |
| C57BL/6 (*M. musculus*) | Bred in-house | C57BL/6 |
| *L. major* (MHOM/JL/80/Friedlin) | Mottram lab (University of York) | N/A |
| *L. mexicana* (MNYC/BZ/62/M379) | Mottram lab (University of York) | N/A |
| **Recombinant DNA** | | |
| pET28a(+) | Novagen | Cat# 69864 |
| Additional plasmids used or generated in this study | Mottram lab (University of York) | See Appendix Table S2 |
| **Antibodies** | | |
| Rabbit anti-HA-tag polyclonal | Bethyl Laboratories | Cat# A190-108A |
| Mouse anti-Myc-tag, clone 4A6, monoclonal | Merck | Cat# 05-724<br>RRID: AB_11211891 |
| Rabbit anti-RPTOR1 polyclonal | Mottram lab (University of York) | N/A |
| Sheep anti-OPB polyclonal | Mottram lab (University of York) | N/A |
| Rabbit anti-Strep-tag II polyclonal | Abcam | Cat# ab76949<br>RRID: AB_1524455 |
| Rabbit anti-SHERP polyclonal | Walrad lab (University of York) | N/A |
| Goat anti-rabbit IgG Alexa Fluor 647 | Thermo Fisher Scientific | Cat# A-21244<br>RRID: AB_2535812 |
| Donkey anti-sheep IgG Alexa Fluor 647 | Thermo Fisher Scientific | Cat# A21448<br>RRID: AB_2535865 |
| Goat anti-rabbit IgG HRP | Promega | Cat# W401B<br>RRID: AB_430833 |
| Goat anti-mouse IgG HRP | Promega | Cat# W402B<br>RRID: AB_430834 |
| **Oligonucleotides and other sequence-based reagents** | | |
| Diagnostic primers and primers used to generate parasite lines | This study | See Appendix Table S1 for details |
| QPCR primers (kDNA minicircles and mouse 5.8S RNA) | (Bezerra-Vasconcelos et al, 2011) (Corrales et al, 2021) | See Appendix Table S1 for details |
| **Chemicals, enzymes and other reagents** | | |
| Q5® High-fidelity DNA polymerase | NEB | Cat# M0491S |
| Rapamycin, mTORC1 complex inhibitor | Abcam | Cat# ab120224 |
| Pierce™ Anti-HA Magnetic Beads | Thermo Fisher Scientific | Cat# 88837 |
| NuPAGE™ 8%, Bis-Tris, 1.0 mm, Midi Protein Gels | Thermo Fisher Scientific | Cat# WG1002BOX |
| NuPAGE™ 4-12%, Bis-Tris, 1.0 mm, Midi Protein Gels | Thermo Fisher Scientific | WG1401BOX |
| G418 (Geneticin) | Invivogen | Cat# ant-gn-1 |
| Hygromycin B Gold | Invivogen | Cat# ant-hg-1 |
| Puromycin | Invivogen | Cat# ant-pr-1 |
| Blasticidin | Invivogen | Cat# ant-bl-1 |
| Ampicillin | Sigma-Aldrich | Cat# A9518-25g |
| DAPI Fluoromount-G | Southern-Biotech | Cat# 0100-20 |
| DAPI | Thermo Fisher Scientific | Cat# D1306 |
| DSP (dithiobis(succinimidyl propionate)), Lomant's Reagent | Thermo Fisher Scientific | Cat# 22585 |
| Protease Inhibitor Cocktail (previously named Proteoloc) | Abcam | Cat# ab271306 |

**Table 1.** (continued)

| Reagent/Resource | Reference or Source | Identifier or Catalog Number |
|---|---|---|
| cOmplete Protease Inhibitor Tablets | Roche/Merck | Cat# 11697498001 |
| cOmplete™ ULTRA Tablets, Mini, EASYpack Protease Inhibitor Cocktail | Roche/Merck | Cat# 05892970001 |
| PhosSTOP | Roche/Merck | Cat# 906845001 |
| EDTA (Ethylenediaminetetraacetic acid) | Sigma-Aldrich | Cat# 60-00-4 |
| MagStrep XT beads | Iba | Cat# 2-4090-002 |
| ProteaseMax | Promega | Cat# V2071 |
| Triton X-100 | Sigma-Aldrich | Cat# 9002-93-1 |
| D-Biotin | Sigma-Aldrich | Cat# B4639 |
| LysoTracker™ Red DND-99 | Thermo Fisher Scientific | Cat# L7528 |
| FM 4-64 | Thermo Fisher Scientific | Cat# T13320 |
| Hoechst 33342 Trihydrochloride, Trihydrate | Thermo Fisher Scientific | Cat# H1399 |
| MEM Amino acids (50x) | Gibco/ Thermo Fisher Scientific | Cat# 11130051 |
| HBSS (10x) | Gibco/Thermo Fisher Scientific | Cat# 14065056 |
| CyGEL | Abcam | ab109204 |
| CyGel Sustain | Abcam | ab109205 |
| Propidium Iodide | Sigma-Aldrich | Cat# P4170 |
| Adenine hemisulfate salt | Sigma-Aldrich | Cat# A9126 |
| 6-Biopterin | Sigma-Aldrich | Cat# B2517 |
| Hemin | Sigma-Aldrich | Cat# H5533 |
| MultiSite Gateway® ThreeFragment Vector Construction Kit | Thermo Fisher Scientific | Cat# 12537-023 |
| DNeasy Blood & Tissue Kit | Qiagen | Cat# 69504 |
| Click-iT AHA Alexa Fluor 488 protein synthesis HCS assay | Thermo Fisher Scientific | Cat# C10289 |
| Click-iT EdU Alexa Fluor 647 Flow Cytometry Assay Kit | Thermo Fisher Scientific | Cat# C10424 |
| ProLong Diamond Antifade Mountant with DAPI | Thermo Fisher Scientific | Cat# P3696 |
| NEBuilder® HiFi DNA Assembly Master Mix | NEB | Cat# E2621L |
| Human T cell Nucleofector Kit | Lonza Bioscience | Cat# VPA-1002 |
| Fast SYBR™ Green Master Mix | Applied Biosystems, Thermo Fisher Scientific | Cat# 4385612 |
| PCRBIO Ultra Mix Red | PCR Biosystems | Cat# PB10.33 |
| Q5 High-Fidelity DNA Polymerase | NEB | Cat# M0491 |
| **Software** | | |
| GraphPad Prism v9 | GraphPad | https://www.graphpad.com/scientific-software/prism/ |
| Xcalibur Software 4.0 | Thermo Fischer scientific | Cat# OPTON-30965 |
| Progenesis QI v2.2 | Waters | https://www.nonlinear.com/progenesis/qi-for-proteomics/ |
| Mascot Daemon v2.6.0 | Matrix Science | https://www.matrixscience.com/mascot_support.html |
| Mascot Server v2.7.0 | Matrix Science | https://www.matrixscience.com/mascot_support.html |
| FCS express v7 | De novo Software | https://denovosoftware.com/ |
| FlowJo | Tree star software | https://www.flowjo.com/ |
| TriTrypDB | (Aslett et al, 2010) | http://tritrypdb.org/tritrypdb/ |
| SAINTq | (Teo et al, 2016) | https://saint-apms.sourceforge.net/Main.html |
| Clustal Omega 3 | (Sievers et al, 2011) | https://www.ebi.ac.uk/Tools/msa/clustalo/ |
| ALINE | (Bond and Schuttelkopf, 2009) | https://bondxray.org/software/aline.html |

**Table 1.** (continued)

| Reagent/Resource | Reference or Source | Identifier or Catalog Number |
|---|---|---|
| TopDraw | (Bond, 2003) | https://www.ccp4.ac.uk/html/topdraw.html |
| STRIDE | (Heinig and Frishman, 2004) | http://webclu.bio.wzw.tum.de/stride/ |
| AlphaFold Monomer v2.0 | Deepmind | https://www.deepmind.com/open-source/alphafold |
| UCSF Chimera v1.14 | UCSF Resource for Biocomputing, Visualization, and Informatics | https://www.cgl.ucsf.edu/chimera/download.html |
| Fiji (ImageJ) | (Schindelin et al, 2012) | https://www.nature.com/articles/nmeth.2019 |
| CLC Main Workbench | Qiagen | https://www.qiagen.com/us/products/discovery-and-translational-research/next-generation-sequencing/informatics-and-data/analysis-and-visualization/clc-main-workbench/ |
| NEBuilder Assembly Tool | NEB | https://nebuilder.neb.com/#!/ |
| QuantStudio Design and Analysis cloud software | Thermo Fisher Scientific | https://www.thermofisher.com/uk/en/home/technical-resources/software-downloads/quantstudio-3-5-real-time-pcr-systems.html |
| Zen (Black Edition) | Zeiss | https://www.micro-shop.zeiss.com/en/us/softwarefinder/software-categories/zen-black/ |
| Zen Lite (Blue edition) | Zeiss | https://www.zeiss.com/microscopy/en/products/software/zeiss-zen-lite.html |
| Microvolution® Deconvolution Software | Microvolution | https://www.microvolution.com/ |
| **Other** | | |
| Nucleofector 2b Device | Lonza Bioscience | |

## Methods and protocols

### Mice

Female BALB/c mice, 4–6 weeks old, purchased from Charles River Laboratories were used for animal studies at the Biological Services Facility at the University of York (Heslington, York, UK). Mice were maintained under specified pathogen free conditions in individually ventilated cages with food and water ad libitum and a 12 h light/12 h dark photoperiod in rooms maintained at 56% humidity, 20–21 °C. The studies were carried out in accordance with the Animal (Scientific Procedures) Act 1986 and under UK Home Office regulations using Project License 60/4442. Protocols and procedures were approved by the relevant ethics committees at the University of Glasgow and the University of York (University of York Animal Welfare and Ethical Review Board). Sample size was determined based on minimum number of animals required for statistics using a power calculation (significance level at 5% and power at 80%) using trial data from 3 mice (mean = 3, standard deviation = 1, difference of 30%). A total of 9 animals were required per group. Blinding was not possible, but animals were randomly selected for groups. No animals were excluded for analysis. Outcome measures assessed were ear lesion score (primary outcome measure) and parasite burden (secondary outcome measure).

### Leishmania parasites

*L. major* (MHOM/JL/80/Friedlin) and *L. mexicana* (MNYC/BZ/62/M379) were grown as promastigotes in HOMEM medium (modified Eagle's medium, Invitrogen) supplemented with 10% (v/v) heat-inactivated foetal calf serum (Gibco) and 1% Penicillin/Streptomycin solution (Sigma-Aldrich) at 25 °C. Transgenic parasite lines were cultured in the presence of appropriate antibiotics at the following concentrations: 50 µg ml$^{-1}$ puromycin, 50 µg ml$^{-1}$

hygromycin B, 10 µg ml$^{-1}$ blasticidin and 25 µg ml$^{-1}$ G418 (all from Invivogen).

## Generation of cell lines

Inducible RPTOR1 null mutant (*RPTOR1$^{-/flox}$*) lines were generated using a modified approach (Strategy 2 in (Duncan et al, 2019)) of the DiCre inducible system (Duncan et al, 2016) (Fig. EV2A). Plasmids were designed with CLC Main Workbench (Qiagen), constructed using Gateway cloning and transfected into log-stage *Leishmania major* promastigotes using a Human T cell Nucleofector Kit (Lonza Bioscience) as previously described (Duncan et al, 2016). Briefly, a stable DiCre expressing cell line (DiCre) was generated by integrating genes for the two subunits of Cre-recombinase fused to FK506-binding protein (FKBP12) and the binding domain of the FKBP12-rapamycin associated protein into the 18S ribosomal RNA locus in *L. major* (Fig. EV2). This DiCre line was then used to generate an inducible RPTOR1 null mutant (*RPTOR1$^{-/flox}$*) by replacing the first *RPTOR1* allele (LmjF.25.0610) with a LoxP flanked (floxed) C-terminal GFP fused *RPTOR1* gene and replacing the second allele with a Hygromycin resistance cassette. Wildtype (WT) and C269A mutant RPTOR1 re-expression (complementation) plasmids and required oligonucleotides were designed using the NEBuilder Assembly Tool (NEB). Plasmids were then generated by Gibson assembly using NEBuilder® HiFi DNA Assembly Master Mix (NEB) and site-directed mutagenesis and transfected into an inducible *RPTOR1$^{-/flox}$* null mutant (*RPTOR1$^{-/flox}$*) line (cl2). For mass spectrometry analyses, *RPTOR1* (LmxM.25.0610), *TOR1* (LmxM.36.6320) and the control bait LmxM.29.3580 were N-terminally endogenously Twin Strep-tagged via CRISPR-Cas9 editing as previously described (Beneke et al, 2017). The control bait was chosen due to its cytoplasmic localisation and provides a more stringent control compared to a

blank affinity purification (Baker et al, 2021). A new pPLOTv1 Twin Strep::mNG::Twin Strep plasmid (pGL2921) which was generated in this study from pPLOTv1 puro-mCherry-puro (Beneke et al, 2017) was used as template for donor DNA. For co-immunoprecipitation, *TOR1* (LmxM.36.6320) was N-terminally endogenously Myc-tagged in *L. mexicana* Cas9 T7 promastigotes using the same CRISPR-Cas9 editing approach; pPLOTv1 blast-mNeonGreen-blast was used as template for donor DNA. This latter line and the *L. mexicana* Cas9 T7 line were then transfected with the WT RPTOR1-HA complementation plasmid to generate single or dual tagged lines. For localisation, both alleles of endogenous *RPTOR1* (LmxM.25.0610) was C-terminally mNeonGreen-myc-tagged in *L. mexicana* Cas9 T7 promastigotes using CRISPR-Cas9 editing from pPLOTv1 puro-mNeonGreen-puro and pPLOTv1 blast-mNeonGreen-blast (Beneke et al, 2017) to generate RPTOR1::mNG. Oligonucleotides and plasmids used in this study are summarised in Appendix Table S1 and Appendix Table S2.

## Affinity purification of Strep-RPTOR1 and Strep-TOR1

Parasites expressing Strep-RPTOR1, Strep-TOR1 or the control bait LmxM.29.3580 were cultured to a density of $7.5 \times 10^6$ parasites $ml^{-1}$. $7.5 \times 10^8$ parasites per biological replicate were harvested by centrifugation for 10 min at $1200 \times g$ and washed twice in PBS. Parasites were re-suspended at a density of $7.5 \times 10^7$ parasites $ml^{-1}$ in pre-warmed PBS. Dithiobis(succinimidyl propionate) (DSP, Thermo Fisher) cross-linker was added to a final concentration of 1 mM and cross-linking proceeded for 10 min at 26 °C. DSP was quenched by adding Tris-HCl to 50 mM and incubating for 10 min shaking at RT. Parasites were centrifuged for 3 min at $1200 \times g$ and frozen at $-80$ °C. Each parasite pellet was lysed in 400 µL lysis buffer (1% IgePal-CA-630, 50 mM Tris pH 7.5, 250 mM NaCl, 1 mM EDTA, 0.1 mM PMSF, 1 µg $ml^{-1}$ pepstatin A, 1 µM E64, 0.4 mM 1-10 phenanthroline). Every 10 ml of lysis buffer was additionally supplemented with 200 µl Proteoloc protease inhibitor cocktail containing w/v 2.16% 4-(2-aminoethyl)benzenesulfonyl fluoride hydrochloride, 0.047% aprotinin, 0.156% bestatin, 0.049% E-64, 0.084% Leupeptin, 0.093% Pepstatin A (Abcam), 3 tablets complete protease inhibitor EDTA free (Roche) and 1 tablet PhosSTOP (Roche). Parasites were lysed by sonication with a microtip sonicator on ice for 3 rounds of 10 s each at an amplitude of 30. Insoluble material was pelleted by centrifugation at $10,000 \times g$ for 10 min at 4 °C. The cleared supernatant was added to 50 µL of MagStrep XT resin and baits were affinity purified with end-over-end rotation for 2 h at 4 °C. Resin was washed 4×, using 300 µL of ice-cold lysis buffer for each wash, followed by 2× washes with 300 µL ice cold PBS. Bait proteins were eluted in two rounds with 25 µL 50 mM biotin in 50 mM TEAB for 10 min each round. Proteins were precipitated with addition of 200 µL methanol then 50 µL chloroform. After vortex mixing, proteins were pelleted by centrifugation at $18,000 \times g$ for 1 h at 4 °C. The protein pellet was washed with ice cold methanol then resuspended in 200 µL 0.01% ProteaseMax in 50 mM TEAB, 10 mM TCEP, 10 mM IAA and 1 mM $CaCl_2$. 200 ng of trypsin/lys-C (Promega) was added and proteins were digested overnight at 37 °C. Digests were acidified by adding TFA to 0.5% and incubated at RT for 1 h. After clarifying digests at $18,000 \times g$ for 10 min, peptides were desalted using in house made C18 StageTips.

## Mass spectrometry data acquisition

Peptides were re-suspended in aqueous 0.1% trifluoroacetic acid (v/v) then loaded onto an mClass nanoflow UPLC system (Waters) equipped with a nanoEaze M/Z Symmetry 100 Å $C_{18}$, 5 µm trap column (180 µm × 20 mm, Waters) and a PepMap, 2 µm, 100 Å, $C_{18}$ EasyNano nanocapillary column (75 mm × 500 mm, Thermo). The trap wash solvent was aqueous 0.05% (v:v) trifluoroacetic acid and the trapping flow rate was 15 µL/min. The trap was washed for 5 min before switching flow to the capillary column. Separation used gradient elution of two solvents: solvent A, aqueous 0.1% (v:v) formic acid; solvent B, acetonitrile containing 0.1% (v:v) formic acid. The flow rate for the capillary column was 300 nL/min and the column temperature was 40 °C. The linear multi-step gradient profile was: 3–10% B over 7 min, 10–35% B over 30 min, 35–99% B over 5 min and then proceeded to wash with 99% solvent B for 4 min. The column was returned to initial conditions and re-equilibrated for 15 min before subsequent injections.

The nanoLC system was interfaced with an Orbitrap Fusion Tribrid mass spectrometer (Thermo) with an EasyNano ionisation source (Thermo). Positive ESI-MS and $MS^2$ spectra were acquired using Xcalibur software (version 4.0, Thermo). Instrument source settings were: ion spray voltage, 2,100 V; sweep gas, 0 Arb; ion transfer tube temperature; 275 °C. $MS^1$ spectra were acquired in the Orbitrap with: 120,000 resolution, scan range: *m/z* 375–1500; AGC target, $4e^5$; max fill time, 100 ms. Data dependent acquisition was performed in top speed mode using a 1 s cycle, selecting the most intense precursors with charge states >1. Easy-IC was used for internal calibration. Dynamic exclusion was performed for 50 s post precursor selection and a minimum threshold for fragmentation was set at $5e^3$. $MS^2$ spectra were acquired in the linear ion trap with: scan rate, turbo; quadrupole isolation, 1.6 *m/z*; activation type, HCD; activation energy: 32%; AGC target, $5e^3$; first mass, 110 *m/z*; max fill time, 100 ms. Acquisitions were arranged by Xcalibur to inject ions for all available parallelizable time.

## Mass spectrometry data analysis

Peak lists in .raw format were imported into Progenesis QI (Version 2.2., Waters) and LC-MS runs aligned to the common sample pool. Precursor ion intensities were normalised against total intensity for each acquisition. A combined peak list was exported in .mgf format for database searching against the *L. mexicana* subset of the TriTrypDB database (8250 sequences; 5,180,224 residues), appended with common proteomic contaminants (116 sequences; 38,371 residues). Mascot Daemon (version 2.6.0, Matrix Science) was used to submit the search to a locally running copy of the Mascot program (Matrix Science Ltd., version 2.7.0). Search criteria specified: Enzyme, trypsin; Max missed cleavages, 1; Fixed modifications, Carbamidomethyl (C); Variable modifications, Oxidation (M), Phosphorylation (S,T); Peptide tolerance, 3 ppm; MS/MS tolerance, 0.5 Da; Instrument, ESI-TRAP. Peptide identifications were passed through the percolator algorithm to achieve a 1% false discovery rate assessed against a reverse database and individual matches filtered to require minimum expect score of 0.05. The Mascot .XML result file was imported into Progenesis QI and peptide identifications associated with precursor peak areas and matched between runs. Relative protein abundance was calculated using precursor ion areas from non-conflicting unique

peptides. Accepted protein quantifications were set to require a minimum of two unique peptide sequences. Statistical testing was performed in Progenesis QI from ArcSinh normalised peptide abundances and ANOVA-derived p-values were converted to multiple test-corrected q-values using the Hochberg and Benjamini approach. Label free protein intensities were analysed with SAINTq (Teo et al, 2016) to determine interacting proteins. Identified proteins in RPTOR1 and TOR1 affinity purifications were quantified relative to levels in the control bait (LmxM.29.3580) purification. Prey scores were filtered to achieve an overall false discovery rate of 10% in the final list of interactors.

## Functional annotation of genes identified by mass spectrometry

Proteins identified in the proteomic analyses were further annotated by homology-based methods and searches in TriTrypDB (Amos et al, 2022; Aslett et al, 2010). A BLASTp (Altschul et al, 1997) search was performed against the NCBI Genbank non-redundant (NR) sequence database(Pruitt et al, 2005) with an E-value threshold $< = 1e-05$.

## Co-immunoprecipitation

Co-immunoprecipitation was performed using TOR1-Myc and RPTOR1-HA dual or single tagged log-stage promastigotes. $7.5 \times 10^8$ parasites were centrifuged at $1200 \times g$ for 10 min, resuspended in 1 mL cold PBS and again centrifuged at $1200 \times g$ for 5 min. Cells were then resuspended in PBS, 2 mM DSP was added and cells were incubated for 10 min shaking at RT. This was followed by adding 50 mM Tris-Cl (final) pH-8 and incubating for 10 min shaking at RT. Cells were pelleted by centrifugation and resuspended in lysis buffer (20 mM Tris-Cl pH 8, 150 mM, NaCL, 0.5% NP-40, 2X mini complete Ultra (Roche), 0.5 mM EDTA, 10 µM E-64. The cells were sonicated and lysed at 3× 30 s 5 W output on ice. The insoluble material was removed by centrifuging at $16,000 \times g$ for 10 min at 4 °C and the supernatant was transferred into a fresh tube. 30 µL of HA Dynabeads (Thermo Fischer) were transferred to a 1.5 mL tube and washed 3× in lysis buffer using a magnetic rack for 2 min each wash. The cell extract was transferred on the beads and incubated for 1 h at 4 °C under rotation. The beads were washed six times in 1 mL of wash buffer on a magnetic rack.

## Antibodies and western blotting

Polyclonal anti-RPTOR1 antibodies were raised in rabbits using an *L. major* recombinant RPTOR1 fragment consisting of the N-terminal 462 amino acids of the protein. The recombinant fragment was obtained by amplifying the relevant sequence from *L. major* genomic DNA, cloning the DNA fragment into pET28a(+) (Novagen) using oligonucleotides described in Appendix Table S1, and transforming into *Escherichia coli* BL21 DE3 (pLysS). The cells were then grown in LB medium containing 37 µg ml⁻¹ chloramphenicol and 20 µg ml⁻¹ kanamycin, until an A600 of 0.6 was reached and then induced with 1 mM isopropyl-β-d-thiogalactopyranoside overnight at 20 °C. The resulting 49 kDa protein was used for antibody generation in rabbits and the antibodies were affinity purified of from the rabbit sera.

For western blotting to confirm complementation lines, $1 \times 10^7$ rapamycin-induced cells were harvested and washed once in PBS by centrifugation at $1200 \times g$. Protein samples were prepared by lysing the pellet of parasites in laemmli buffer and boiling for 5 min; $5 \times 10^6$ parasites were loaded per well. For co-immunoprecipitation and affinity purification samples, $7.5 \times 10^8$ cells were used as starting material and processed as described in the relevant method sections above. Input samples (20 µg for immunoprecipitations and $2.25 \times 10^7$ cells for affinity purification per line), taken before addition of HA beads or MagStrep XT resin, were prepared in laemmli buffer and heated as specified for samples below. For HA-tag immunoprecipitations proteins were eluted from beads in 50 µl of laemmli buffer after 6 washes, boiled for 5 min and 25 µl sample was loaded on an 8% NuPAGE Bis-Tris gel (Thermo scientific). For affinity purification proteins were eluted by adding 25 µl 50 mM biotin 50 mM TEAB to the MagStrep XT resin, eluting for 10 mins shaking at 700 rpm at RT. Then 16.6 µl of laemmli buffer was added to this eluate, heated at 70 °C for 10 mins and 20 µl loaded on a 4–12% NuPAGE Bis-Tris gel (Thermo Fisher Scientific). Each gel was transferred to a PVDF membrane by wet transfer. The membrane was blocked in 5% milk for 1 hr, followed by incubation with rabbit anti-HA antibodies (1:3000, Bethyl Laboratories), chicken anti-Myc antibodies (1:3000, Bethyl Laboratories), rabbit anti-RPTOR1 antibodies (1:200), sheep anti-OPB antibodies (1:20,000) or rabbit anti-strep-tag-II antibodies (1:2000, Abcam ab76949). The membrane was washed 3 times and incubated with HRP-conjugated secondary antibodies (1:5000). The blots were developed using the SuperSignal West Pico substrate (Thermo Fisher).

## Localisation by live cell microscopy

RPTOR1::mNG *L. mexicana* procyclic promastigotes were treated with FM 4-64 to label endocytic compartments and the lysosome or Lysotracker Red DND-99 (both from Thermo Fisher Scientific) to label acidocalcisomes. For FM4-64 labelling, ~$1.5 \times 10^6$ cells were incubated with 20 µM FM 4-64 (from a 10 mM stock in DMSO) in 500 µL Homem at 28 °C for 15 min, then washed by centrifuging at $1200 \times g$, resuspended in fresh HOMEM media and incubated for another 45 min at 28 °C. 10 µg mL⁻¹ of Hoechst 33342 was added for the last 10 min of incubation. Cells were then washed in HOMEM media, resuspended in 5 µL HOMEM and mixed with 95 µL Cygel Sustain (Abcam) that had been primed with 10× HBSS supplemented with MEM amino acids (both from Thermo Scientific). Samples were transferred to slides on ice, covered with a coverslip and imaged on a 63× oil immersion Plan Apochromat Differential Interference Contrast (DIC) objective on a Zeiss LSM880 confocal microscope. Imaging was performed with the following excitation and emission wavelengths: Hoechst at 405 nm excitation and 415–490 nm emission wavelengths, mNeonGreen with 488 nm excitation and 490–544 nm emission wavelengths, FM 4-64 with 561 nm excitation and 645-758 nm emission wavelengths. Zen Black and Zen Lite Imaging software was used for image capturing and processing, respectively. For Lysotracker labelling, cells were incubated with 100 nM Lysotracker Red DND-99 in HOMEM media at 28 °C for 1 h, washed with PBS, resuspended in 10 µg mL⁻¹ of Hoechst 33342 (Thermo Fisher Scientific) in PBS and incubated for 10 min at room temperature protected from light. Cells were centrifuged and the pellet was resuspended in 50 µL of ice chilled CyGEL™ (Abcam). Samples were transferred to slides, covered with a coverslip and immediately imaged using a 63× oil

immersion DIC II Plan Apochromat objective on a Zeiss AxioObserver Inverted Microscope with a Colibri 7 narrow-band LED system and white LED. Imaging was performed with the following light sources and filters: Hoechst with the 385 nm LED and 430–470 nm emission filter, mNeonGreen with the 475 nm LED and 500–550 nm emission filter, Lysotracker Red with the 567 nm LED and 570–640 nm emission filter. Image Z-stacks were blind deconvolved with the Microvolution plugin for ImageJ (Fiji plugin) (Schindelin et al, 2012) using 100 iterations.

## Induction of DiCre mediated *RPTOR1* deletion

In this DiCre system gene excision can be induced by addition of 100 nM to 1 µM rapamycin (Abcam), which dimerizes the Cre subunits resulting in an active diCre recombinase. Mid-log stage promastigotes were diluted to a density of $1 \times 10^5$ cells mL$^{-1}$ and induced for 3 days by daily addition of 100 nM rapamycin (Abcam) in DMSO, or DMSO alone (0.1%) in control cultures. After this initial induction, cells were counted and diluted in fresh supplemented HOMEM medium at a density of $1 \times 10^5$ cells mL$^{-1}$ with daily addition of 100 nM rapamycin until harvested for analysis.

## Diagnostic PCR to assess for presence of transgenes and *RPTOR1*

$1–5 \times 10^6$ cells were centrifuged at $1000 \times g$ for 8 min, washed once in PBS and frozen at $-20\,°C$. DNA was extracted using the QIAGEN DNeasy Blood and Tissue Kit according to the manufacturer's instructions for animal cells. PCR was performed using Q5 High-Fidelity DNA polymerase according to the manufacturer's instructions. All oligonucleotides used in this study are summarised in Appendix Table S1.

## Flow cytometry to analyse cell size, RPTOR1-GFP expression, viability and cell cycle

At the times indicated in figure legends cells were prepared for flow cytometry. Briefly, $5 \times 10^6$ cells were centrifuged at $1000 \times g$ for 8 min and washed twice in PBS with 5 mM EDTA (PBS/EDTA). To assess cell size, viability, and RPTOR1-GFP expression, live cells were then resuspended in 1 mL of PBS/EDTA with 1 µg mL$^{-1}$ of propidium iodide (PI) to allow assessment or exclusion of dead cells. For cell cycle analysis cells were fixed in 70% methanol in PBS/EDTA at 4 °C for 1 h or overnight and washed twice in PBS/EDTA by centrifugation at $1000 \times g$ for 5 min. Cells were resuspended in 1 mL of PBS/EDTA with 10 µg mL$^{-1}$ of PI and 10 µg mL$^{-1}$ of RNAseA, and incubated at 37 °C for 45 min. Fixed or live cells were analysed on a Beckman Coulter, CyAn ADP and data were analysed FCSExpress v7 (De novo software).

## Cell proliferation and clonogenic survival assay

To assess proliferation, cells were counted daily using a hemocytometer or Z1 Beckman Coulter counter before and after rapamycin induction. For the clonogenic survival assay, cells were induced for 72 h with rapamycin or DMSO, counted and diluted to 1.6 parasites mL$^{-1}$ in HOMEM with 20% heat-inactivated FCS and 100 nM

rapamycin or DMSO. The cells were plated out in two to four 96-well flat bottom plates by adding 200 µL cells per well. Plates were sealed and incubated at 25 °C for 3–4 weeks. Surviving clones were counted by visual inspection of each well using a light microscope; any well containing live parasites was counted as a surviving clone. The percentage of surviving clones of the total cells plated is shown in graphs.

## Protein synthesis

Protein synthesis was assessed using a Click-iT AHA Alexa Fluor 488 Protein Synthesis HCS Assay (Thermo Fisher, Cat# C10289). Click-iT reagents were prepared according to the manufacturer's instructions. Cells were centrifuged at $1000 \times g$ for 8 min and washed once in methionine-free RPMI medium (Thermo Fisher) supplemented with 10% of 3.5 kDa-dialysed heat inactivated FBS, 1 M HEPES (pH 7.4), 5 mM Adenine (Sigma A9126 Adenine Hemisulfate salt), 0.25% (2.5 mg ml$^{-1}$) Hemin in 50% triethanolamine (Sigma H5533) or in 50 mM NaOH, 200 mM L-glutamine (Gibco), Pen/strep (Gibco) and 0.3 mg mL$^{-1}$ Biopterin (methionine-free medium (MFM). Cells were then resuspended in MFM containing 50 µM Click-iT® AHA working solution, transferred to 1.5 mL tubes ($3 \times 10^6$ cells per tube) in triplicate and incubated for 1–2 h. After incubation, cells were centrifuged at $1000 \times g$ for 5 min, washed once in PBS and fixed with 1% formaldehyde in PBS for 15 min at room temperature. Cells were then washed twice in 3% BSA in PBS and permeabilized using 0.1% Triton-X100 for 10 min at room temperature. Permeabilized cells were washed twice in 3% BSA in PBS, resuspended in 100 µL Click-iT reaction cocktail and incubated for 30 min at room temperature, protected from light. Cells were then washed once in 3% BSA in PBS and once in PBS alone by centrifugation at $1000 \times g$ for 5 min, and finally transferred to a black 96-well flat-bottomed plate in 100 µL PBS. Alexa-Fluor 488 fluorescence was measured using a Clariostar microplate reader (BMG).

## Sequence alignment and topology diagrams

Sequences of caspase 7 and RPTOR1 from human and *A. thaliana* were obtained from UniProtKB (P55210, Q8N122 and Q93YQ1, respectively). *L. major* and *T. brucei* RPTOR1 sequences were obtained from TriTrypDB (Amos et al, 2022; Aslett et al, 2010). The alignment of primary protein sequences was done using Clustal Omega 3 (Sievers et al, 2011) and ALINE (Bond and Schuttelkopf, 2009); secondary structure alignment was done using PDBeFold with SSM ((Bond and Schuttelkopf, 2009; Krissinel and Henrick, 2004) from the structures of human Caspase_7 (PDB code 1F1J) and *A. thaliana* RAPTOR1 (PDB code 5WBI). STRIDE (Heinig and Frishman, 2004) was used to assign secondary structure in the topology diagrams and the diagrams were produced by manual inspection of the 3D structures in PyMol (DeLano, 2002) and subsequent transfer of the secondary structural elements onto a 2D plane using TOPDRAW (Bond, 2003). The Alphafold model (AF-A4I1A2-F1-model_v3.pdb) of LinfRPTOR1 was last updated in AlphaFold DB version 2022-06-01 and created with the AlphaFold Monomer v2.0 pipeline. AtRAPTOR1 and LinfRPTOR1 were superposed in UCSF Chimera (V1.14) (Pettersen et al, 2004) using the MatchMaker tool (Needleman-Wunsch Algorithm, BLOSUM-62 Matrix and default parameters). The Match->Align tool was

used to generate amino acid alignments from the structural superposition.

## Assessment of metacyclogenesis and retroleptomonad growth

PNA⁻ cells were isolated from cultures by agglutination of promastigotes with $50 \mu g\ ml^{-1}$ peanut lectin as previously described[23]. SHERP expression was measured using flow cytometry after staining with affinity-purified rabbit anti-SHERP (Knuepfer et al, 2001) and goat anti-rabbit AF647 antibodies. Briefly, cells were washed in PBS by centrifugation at $1000 \times g$ for 3 min and fixed in 1% paraformaldehyde in PBS at 4 °C overnight. Cells were washed again in PBS and permeabilized in 0.1% Triton X-100 for 5 min. After another three washes in PBS, cells were blocked for 30 min in PBS containing 10% FCS and 5% goat serum followed by incubation for 1 h with rabbit anti-SHERP antibody (1:100 dilution) in blocking buffer, both on ice. Cells were washed three times in PBS and stained with goat anti-rabbit AF647 in PBS for 30 min on ice. Stained cells were washed twice in PBS and resuspended in PBS with 5 mM EDTA before analysis on a Beckman Coulter, CyAn ADP. Data were analysed on FlowJo software (Tree Star Inc.). To assess growth of retroleptomonads, promastigotes were purified twice using PNA agglutination as described above and resuspended in HOMEM containing 20% FCS followed by daily counting of live cells.

## Macrophage infections

Peritoneal exudate cells were harvested from wild-type C57/BL6 mice that had been injected intraperitoneal with 4% Brewer's Thioglycollate four days before harvest. Two females were used for one experiment (infection ratio 6:1) and two males for a second experiment (infection ratio 3:1) described below. Peritoneal cells were plated on petri-dishes for 3 h and adherent macrophages harvested. Macrophages were replated in DMEM supplemented with 10% FCS, 2 mM L-glutamine, 100 U ml⁻¹ penicillin, 100 μg ml⁻¹ streptomycin (DMEM10) on either sterile glass coverslips in 24-well plates (for EdU incorporation), in 12-well plates (for parasite harvest) or in 16-well Nunc Lab-Tek chamber slides (Thermo Fisher) for use in infection assays and rested overnight. PNA⁻ promastigotes were added to peritoneal macrophages at a 3:1 or 6:1 ratio (specified in figure legends) and washed off after 3 h using warmed DMEM supplemented with 10% FCS, 2 mM L-glutamine, 100 U ml⁻¹ penicillin, 100 μg ml⁻¹ streptomycin. For counting parasites per 100 macrophages 16-well infection slides were washed at 3 h, day 1 and day 4 after infection with PBS, fixed with 2% paraformaldehyde, permeabilized with 100% methanol and stained with DAPI using DAPI Fluoromount-G (SouthernBiotech). Cells were imaged by fluorescence microscopy using a ×63 oil immersion DIC II Plan Apochromat objective on a Zeiss AxioObserver Inverted Microscope with a Colibri 7 narrow-band LED system and white LED. Imaging of DAPI was performed with the 385 nm LED and 430-470 nm emission filter. Analysis was performed using Zen Blue (Zeiss) and ImageJ (Fiji plugin) (Schindelin et al, 2012) image analysis software.

## Analysis of intracellular parasite morphology

At day 5 p.i. infected macrophages were washed with PBS and harvested by scraping wells with Hank's Balanced Salt Solution (HBSS). Macrophages were lysed by incubating at RT with occasional vortexing in 10 ml of 1.25 mg.mL⁻¹ saponin in HBSS for 5 min. Cell were washed three times by centrifuging at $2200 \times g$ for 5 min in 10 mL HBSS. After the final wash cells were resuspended in 1 mL HBSS and unclumped by aspirating in a 1 mL syringe with 27 G needle. Cells were centrifuged at $2200 \times g$ for 5 min, resuspended in 20 μL PBS and added to a poly-L-lysine coated coverslips to settle for 30 min. PNA⁻ promastigotes were added to separate poly-L-lysine coated coverslip to settle for 30 mins. PBS was removed, cells were fixed in 2% PFA for 15 min, washed once in PBS and permeabilized in 0.1% Triton X-100 in PBS for 10 min at RT. Cell were washed twice in PBS, air-dried for 5 min and blocked in blocking buffer (1% BSA/0.1% Triton X-100/ 0.1% cold fish gelatin in PBS) for 1 h at RT. Cell were then stained overnight at 4 °C with sheep anti-OPB primary antibody (1:100) in blocking buffer. Cells were washed twice in PBS and stained with donkey anti-sheep IgG, AF647 (1:500, Invitrogen, Thermo Fisher Scientific) in blocking buffer for 30 min at RT. Cells were washed three times in TBST and stained with DAPI (1 μg.mL⁻¹ in PBS, Thermo Fisher Scientific) for 1 min. Cell were mounted with ProLong Diamond antifade mountant containing DAPI and cured overnight before imaging by fluorescence microscopy using a ×100 oil immersion DIC II Plan Apochromat objective on a Zeiss AxioObserver Inverted Microscope with a Colibri 7 narrow-band LED system and white LED. Imaging was performed with the following light sources and filters: DAPI with the 385 nm LED and 430–470 nm emission filter and OPB with the 630 nm LED and 665–715 nm emission filter. Images were analysed using Zen Blue and ImageJ software (Fiji plugin) (Schindelin et al, 2012).

## Assessment of parasite proliferation by EdU incorporation

EdU incorporation into intracellular parasites was assessed by microscopy using a Click-iT EdU Alexa-Fluor 647 Flow Cytometry Assay Kit (Thermo Fisher Scientific). Briefly, DMEM10 supplemented with 100 μM EdU or DMSO as control was added to infected macrophage plated on glass coverslips at day 2 post-infection and incubated at 37 °C 5% $CO_2$ for 3 days. At day 5 p.i. cells were washed in PBS and fixed in 2% PFA for 15 min using PBS and Click-iT fixative (Component D) in the kit. All steps were performed on coverslips in 24-well plates. Cells were washed twice in PBS and permeabilized for 15 min at RT by adding 1 volume (200 μL) of the kit's Click-iT saponin-based permeabilization and wash reagent. Five volumes (1 mL) of Click-iT cocktail containing CuSO4, AF-647, reaction buffer and PBS was added to cells for 30 min at RT. Cell were then wash with 15 volumes (3 mL) Click-iT saponin-based permeabilization and wash reagent and stained with DAPI (1 μg.mL⁻¹) in PBS for 1 min at RT. Cells were washed in PBS; the coverslip was transferred to a glass slide, mounted with Prolong Diamond antifade mountant containing DAPI (Thermo Fisher Scientific) and cured overnight. Cells were imaged by fluorescence and white light microscopy using a ×100 oil immersion DIC II Plan Apochromat objective on a Zeiss AxioObserver Inverted Microscope with a Colibri 7 narrow-band LED system. Imaging was performed with the following light sources and filters: DAPI with the 385 nm LED and 430–470 nm emission filter and EdU-AF647 with the 630 nm LED and

665–715 nm emission filter. Images were analysed using Zen Blue and ImageJ software (Fiji plugin) (Schindelin et al, 2012). The same sized Region of Interest (ROI) was used to assess EdU signal in the nucleus of all parasites and the adjacent background signal in the host cell.

## Lesion development in mice

Female BALB/c mice (4–6 weeks, Charles River Laboratories) were infected intradermally in the ear with $1 \times 10^5$ PNA$^-$ metacyclic promastigotes in 10 µl PBS. Lesion development was monitored weekly by using the Schuster scoring system (Schuster et al, 2014).

## qPCR to determine parasite load in ears

At 6 wks p.i. mice were sacrificed and whole ears of naïve and infected mice were harvested, split into halves, cut into smaller pieces and digested for 2.5 h in 500 µl of 100 µg.mL$^{-1}$ collagenase D in DMEM/ 2% FCS while shaking at 37 °C. Ear homogenates were teased through a 70 µm cell strainer and washed by centrifuging in 6 mL PBS at $3000 \times g$ for 8 min to remove collagenase. Pellets were resuspended in 500 µL PBS, centrifuged at $3000 \times g$ for 8 min and, after removing supernatant, frozen and stored at $-80$ °C. Control standard samples were prepared by making a dilution series of *L. major* DiCre or RPTOR1$^{-/-}$ promastigotes and adding parasites to naïve BALB/c ears. Genomic DNA was extracted using the QIAGEN DNeasy Blood and Tissue Kit according to the manufacturer's instructions for animal cells, eluted in 100 µL of AE buffer and stored at $-80$ °C. Quantitative PCR was performed using Fast SYBR™ Green Master Mix (Applied Biosystems, Thermo Fisher Scientific) following the manufacturer's instructions. Two nanogram of genomic DNA (template) was added in 20 µL reactions with primers (200 nM each) for *Leishmania* kinetoplast DNA minicircle (Bezerra-Vasconcelos et al, 2011) and mouse 5.8S RNA genes (Corrales et al, 2021) (Appendix Table S1) and the qPCR run on an Applied Biosystems QuantStudio 3 Real-Time PCR System (Thermo Fisher Scientific). Leishmania kDNA cycles to threshold (Ct) values were normalised using the mouse Ct values and quantified relative to a control standard (naive ears spiked with $1 \times 10^5$ parasites) using the QuantStudio Design and Analysis cloud software.

## PCR to assess the presence of RPTOR1 CDS in ear samples

gDNA of naive and infected BALB/c ears were prepared as described in "qPCR to determine parasite load in ears". PCR was performed using PCRBIO Ultra Mix Red (PCR Biosystems) according to the manufacturer's instructions. Diluted (1:10) gDNA was used as template. All oligonucleotides used in this study are summarised in Appendix Table S1.

## Scanning electron microscopy

Rapamycin or DMSO treated cells were fixed in 4% formaldehyde and 2.5% glutaraldehyde in 0.1 M phosphate buffer, pH 7.3 for 30 min. Cells were washed twice in 0.1 M phosphate buffer for 10 min, adhered to poly-L-lysine-coated coverslips and post-fixed in 1% osmium tetroxide for 45 min on ice. Thereafter, they were washed twice in 0.1 M phosphate buffer for 10 min and dehydrated in a graded series of ethanol concentrations (25–100%) for 15 min each. The final 100% ethanol was replaced with two changes of hexamethyldisilazane (HMDS), and cells were left to air-dry in a desiccator overnight. Samples were affixed to SEM stubs, sputter coated with 20 nm of gold-palladium on Polaron SC7640 sputter coater and then imaged using a JEOL JSM 6490LV scanning electron microscope operating at 8 kV accelerating voltage. Images were analysed on ImageJ software (Fiji plugin) (Schindelin et al, 2012).

## Assessment of adenine response

DiCre, *RPTOR1$^{-/flox}$* Cl2 and *RPTOR1* complementation lines were cultured for five days in Grace's insect medium (Sigma-Aldrich) with 10% heat-inactivated FCS (Gibco) and 1% Penicillin/ Streptomycin solution (Sigma-Aldrich) with daily addition of 100 nM rapamycin (Abcam) in 0.1% DMSO to induce *RPTOR1* excision, or 0.1% DMSO alone in control cultures. After the first three days of culture cells were diluted to $1 \times 10^5$ cells ml$^{-1}$ in fresh media with rapamycin or DMSO and cultured for the remaining two days. After treatment, cells were counted, diluted to a density of $2 \times 10^6$ cells ml$^{-1}$ and cultured for another three days in fresh Grace's medium also supplemented with 500 µM adenine or DMSO as a control. The cells were then counted using a Z1 Beckman Coulter counter.

## Statistical analyses

All statistical analyses were performed with Prism (GraphPad Software, La Jolla, CA, USA) using the test specified in the figure legends. Statistically significant differences ($P < 0.05$) are annotated on the graphs using symbols as described in the figure legends.

# Data availability

Complete mass spectrometry data sets and proteomic identifications from this publication are available to download from MassIVE (MSV000090621) [https://doi.org/10.25345/C5S17SX84] and ProteomeXchange (PXD037832).

# Peer review information

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

## Acknowledgements

This work was supported by the Medical Research Council (MRC MR/K019384/1) and Wellcome Trust (200807) to J.C.M. We thank Karen Hogg, Graeme Park, Grant Calder and Karen Hodgkinson for technical advice and support within the Bioscience Technology Facility; the Biological Services Facility for animal husbandry; Nathaniel Jones for generating the structural overlays with the LinfRPTOR1 Alphafold model and critical review of the manuscript; Pegine Walrad for kindly providing the anti-SHERP antibody and Chris Macdonald for providing FM 4-64 (all University of York). We also thank Manuel Saldivia for critical review of the manuscript. The York Centre of Excellence in Mass Spectrometry was created thanks to a major capital investment through Science City York, supported by Yorkshire Forward with funds from the Northern Way Initiative, and subsequent support from EPSRC (EP/K039660/1; EP/M028127/1). UCSF Chimera was developed by the Resource for Biocomputing, Visualization, and Informatics at the University of California, San Francisco, with support from NIH P41-GM103311.

## Author contributions

**Elmarie Myburgh**: Conceptualization; Resources; Data curation; Formal analysis; Supervision; Validation; Investigation; Visualization; Methodology; Writing—original draft; Project administration; Writing—review and editing. **Vincent Geoghegan**: Resources; Data curation; Formal analysis; Validation; Investigation; Visualization; Methodology; Writing—original draft; Writing—review and editing. **Eliza VC Alves-Ferreira**: Formal analysis; Validation; Investigation; Methodology; Writing—review and editing. **Y Romina Nievas**: Resources; Formal analysis; Validation; Investigation; Visualization; Methodology; Writing—review and editing. **Jaspreet S Grewal**: Resources; Formal analysis; Validation; Investigation; Visualization; Methodology; Writing—review and editing. **Elaine Brown**: Validation; Investigation. **Karen McLuskey**: Conceptualization; Resources; Formal analysis; Validation; Investigation; Visualization; Methodology; Writing—review and editing. **Jeremy C Mottram**:

Conceptualization; Resources; Supervision; Funding acquisition; Methodology; Writing—original draft; Project administration; Writing—review and editing.

## Disclosure and competing interests statement

The authors declare no competing interests.

# Expanded View Figures

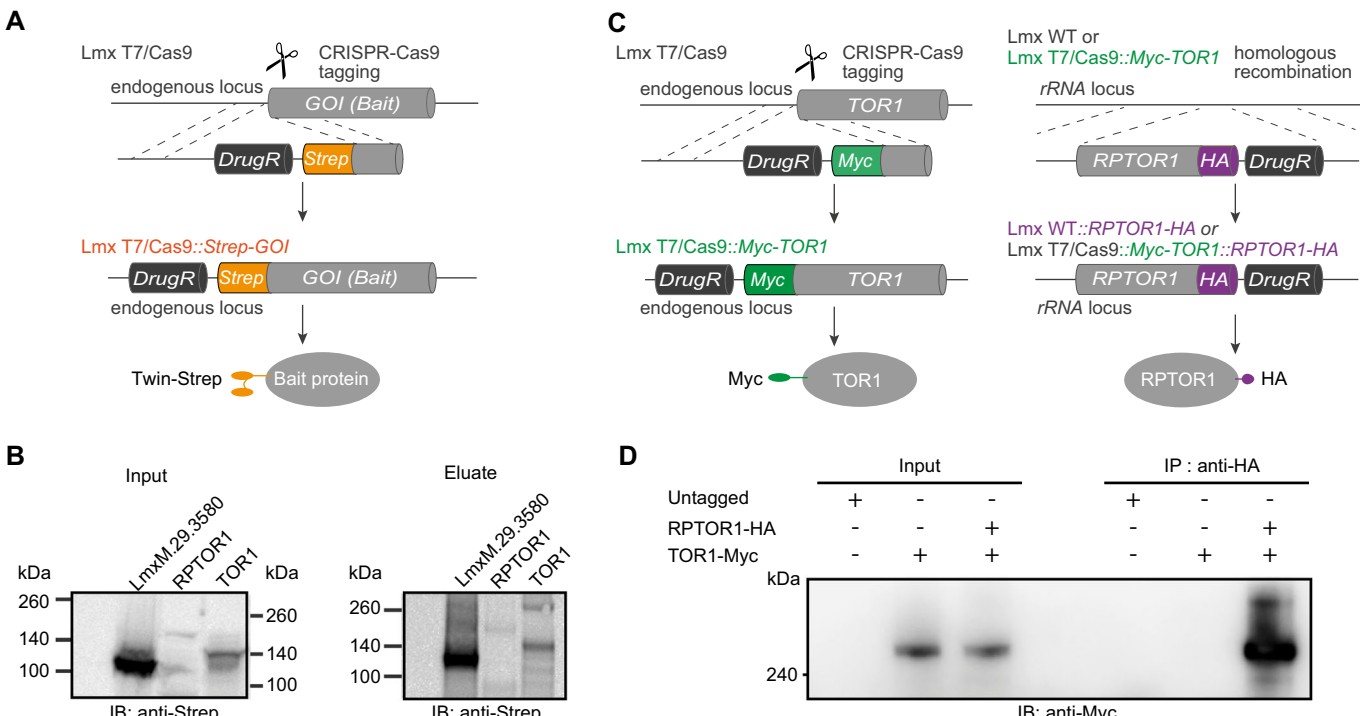

**Figure EV1. RPTOR1 immunoprecipitation strategy.**

(A) Endogenous genes for RPTOR1, TOR1 and control bait LmxM.29.3580 were Twin-Strep-tagged using CRISPR-Cas9 in *L. mexicana* for the affinity purification of these bait proteins and their interactors. GOI: Gene of Interest. (B) A sample of lysate, equivalent to $2.25 \times 10^7$ cells per line from the *L. mexicana* parental line (T7) or Twin-Strep-tagged lines was taken prior to affinity purification (input) and analysed by western blot. Bait proteins were eluted from MagStrep XT resin with biotin and half of the eluate loaded for analysis by western blot (eluate). Predicted sizes are: LmxM.29.3580 85 kDa, RPTOR1 161 kDa, TOR1 291 kDa. IB: Immunoblot. (C) Endogenous *TOR1* was Myc-tagged using a CRISPR-Cas9 tagging approach while HA-tagged *RPTOR1* was inserted in the ribosomal locus using homologous recombination to generate single or dual-tagged *L. mexicana* lines. (D) Lysates of *L. mexicana* expressing untagged RPTOR1 and TOR1, HA-tagged RPTOR1 and/or Myc-tagged TOR1 were incubated with anti-HA-conjugated magnetic beads. Input samples were removed before addition of beads and 20 μg of protein loaded for each sample. After 6 washes, beads were eluted in 50 μL laemmli buffer per line and half of this eluate analysed by western blot using anti-Myc antibodies. IB: Immunoblot. Source data are available online for this figure.

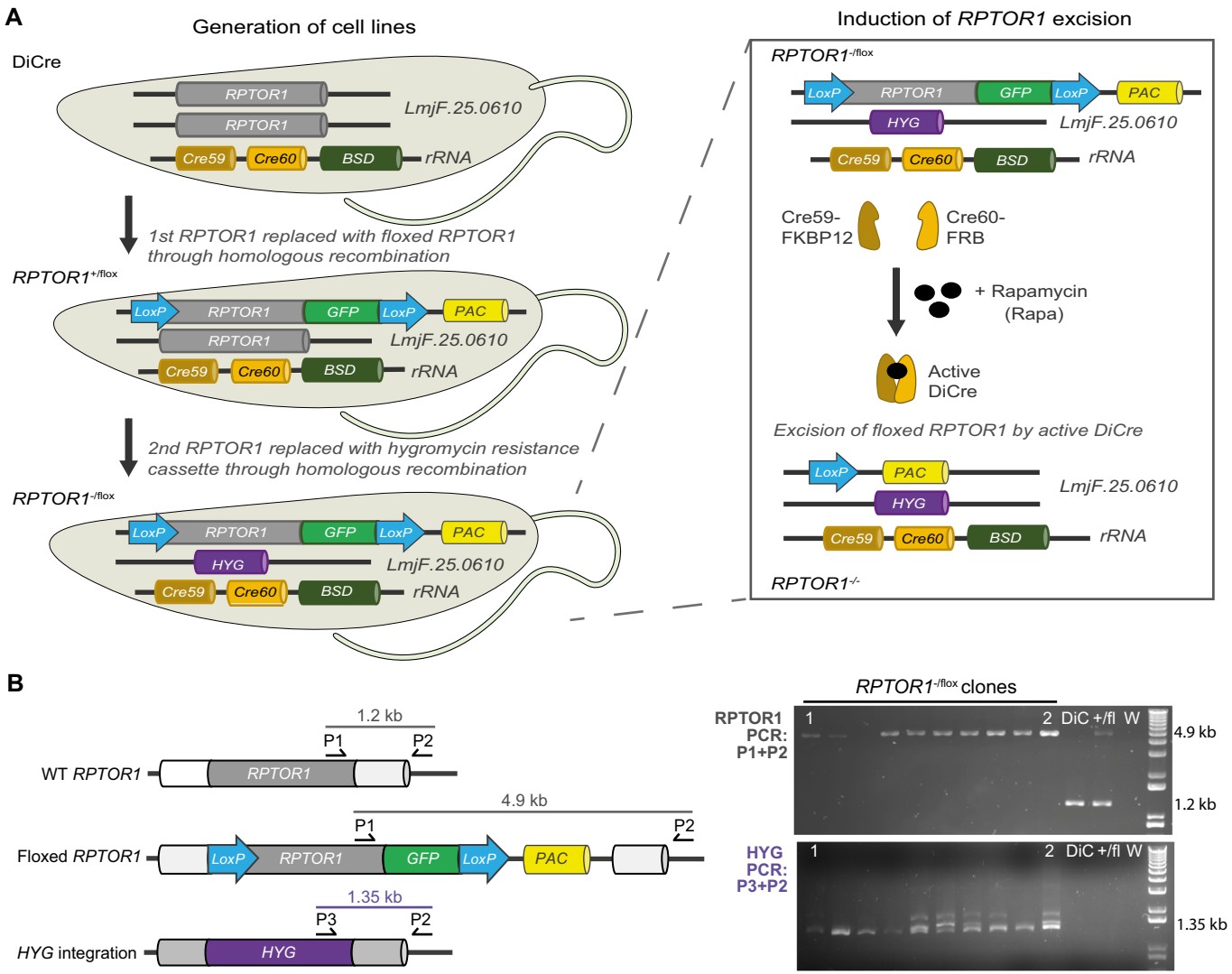

**Figure EV2. RPTOR1 knockout strategy.**

(**A**) Schematic of *RPTOR1* knockout strategy. The background cell line, DiCre, was generated by integrating the diCre expression cassette into the ribosomal RNA locus for constitutive expression of FKBP-Cre59 and FRB-Cre60 in *L. major* Friedlin. The inducible *RPTOR1* knockout line (*RPTOR1⁻/flox*) was generated by replacing the 1st *RPTOR1* allele with a LoxP flanked (floxed) C-terminal GFP-tagged version of *RPTOR1* followed by replacement of the 2nd allele with a hygromycin resistance cassette. The floxed *RPTOR1* gene can be excised by Cre-recombinase following rapamycin induced dimerization to generate a *RPTOR1⁻/⁻* line. (**B**) Diagnostic PCRs of gDNA from generated cell lines confirm integration of hygromycin resistance and floxed *RPTOR1* cassettes. Primer binding site and size of PCR products are shown in the diagram (left). Ten clones of *RPTOR1⁻/flox* are shown after PCR and agarose gel electrophoresis (right); lanes with clones 1 and 2 that are described in this study are indicated on the gel image. DiC, DiCre; +/fl, *RPTOR1⁺/flox*; W, water control.

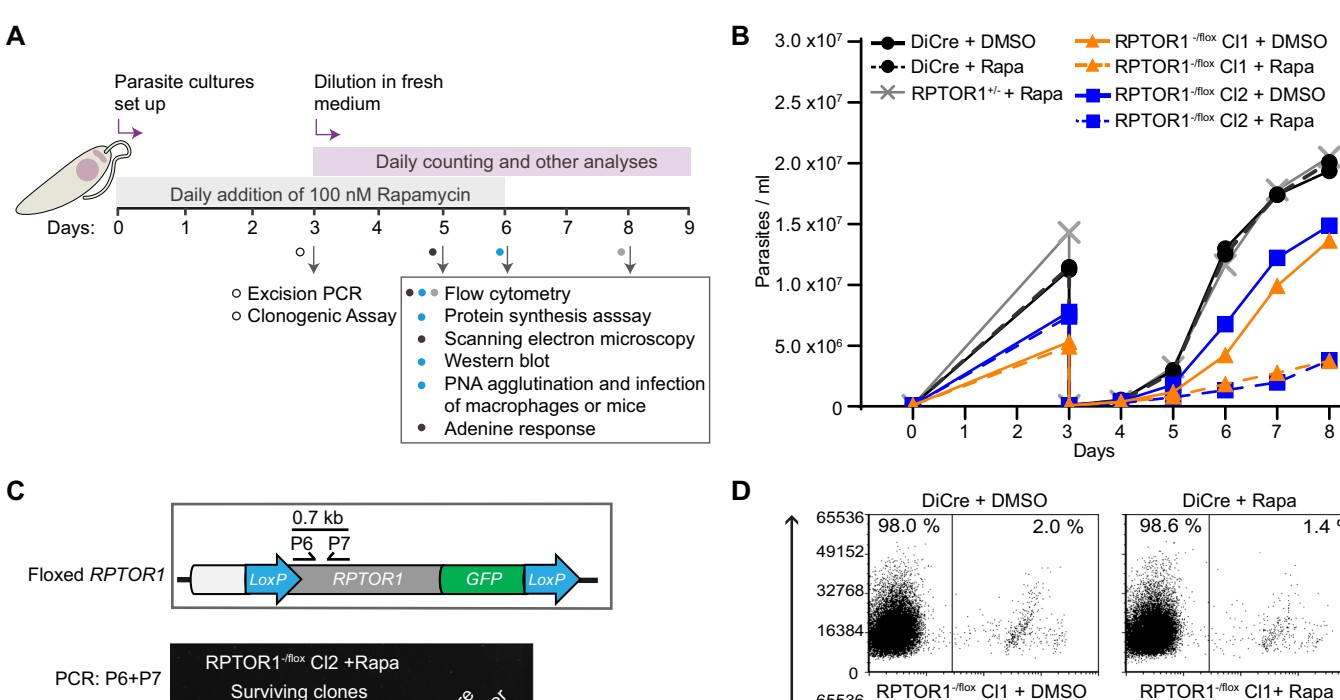

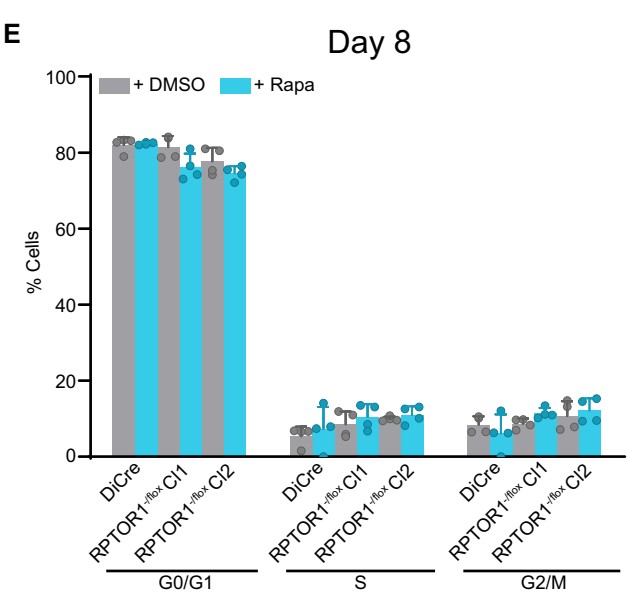

◀ **Figure EV3. RPTOR1 is essential for cell proliferation and long-term survival.**

(A) Schematic of the rapamycin induction and analysis timeline. Log-stage promastigotes were set up at $1 \times 10^5$ cells mL$^{-1}$ (day 0) and treated for three days with daily addition of DMSO or 100 nM rapamycin; cells were then counted and diluted to $1 \times 10^5$ cells mL$^{-1}$ (day 3) followed by culturing for up to 6 additional days for daily counting and other analyses as indicated. (B) Cell densities of uninduced (+DMSO, solid line) and rapamycin-induced (+Rapa, dashed line) cells. Promastigotes of DiCre (black), *RPTOR1*$^{+/-}$ (grey, +Rapa only) and *RPTOR1*$^{-/flox}$ lines, Cl1 (orange) and Cl2 (blue) were counted daily for five days after the initial three days of rapamycin induction. A representative dataset of three to four similar experiments is shown. (C) PCR analysis of genomic DNA from surviving clones (clones 1–7) of rapamycin-induced *RPTOR1*$^{-/flox}$ Cl2 and DiCre cells. Schematic (upper panel) shows the *RPTOR1* locus with floxed *RPTOR1* cassette with primer binding sites and the predicted length of the PCR amplicon. Agarose gel (lower panel) indicates the presence of the *RPTOR1* CDS fragment from the unexcised floxed *RPTOR1* cassette in the seven clones from two clonogenic assays. (D) Cell viability was measured by flow cytometry of propidium iodide-stained cells after five days of induction. Representative dot plots of side scatter versus propidium iodide fluorescence are shown for uninduced (+DMSO) or rapamycin-induced (+Rapa) cells. Numbers indicate the percentages of cells within the gate with live cells shown in the propidium iodide negative (left) gate in each plot. (E) Cell cycle analysis of fixed propidium iodide cells eight days after induction. Data show mean ± SD ($n = 4$ biological replicates) from two experiments; each dot denotes a replicate. Source data are available online for this figure.

**A**

HsCaspase-7

Cys^186

PDB:1F1J

AtRAPTOR1

Cys^245

PDB:5WBI

LinfRPTOR1

Cys^289

Overlay

Overlay

**B**

| | | P | K | L | F | F | I | Q | A | C | R |
|---|---|---|---|---|---|---|---|---|---|---|---|
| HsCaspase-7 | 178 | P | K | L | F | F | I | Q | A | C | R |
| AtRAPTOR1 | 238 | P | S | I | Y | V | F | D | C | S | A |
| LinfRPTOR1 | 262 | P | A | I | Y | V | F | D | C | N | S |

**C**

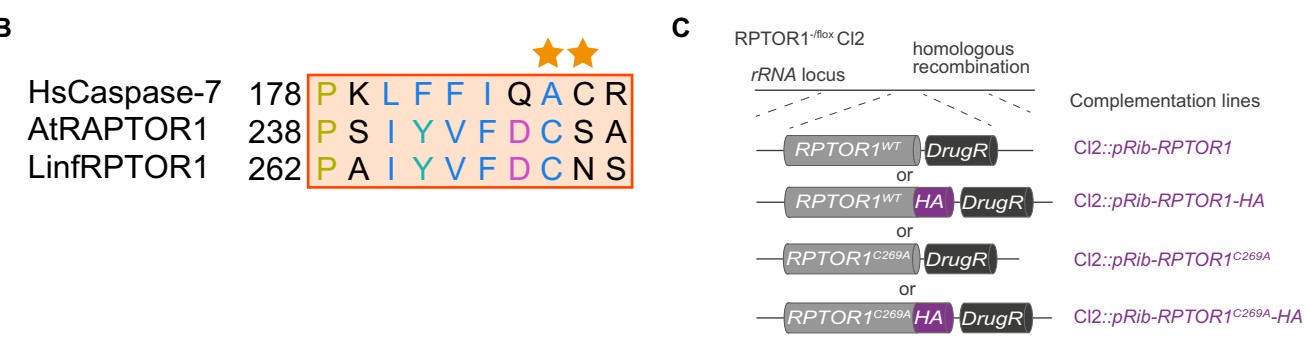

RPTOR1^{-/flox} Cl2

rRNA locus — homologous recombination

Complementation lines

RPTOR1^{WT} — DrugR          Cl2::pRib-RPTOR1

or

RPTOR1^{WT} HA - DrugR       Cl2::pRib-RPTOR1-HA

or

RPTOR1^{C269A} — DrugR       Cl2::pRib-RPTOR1^{C269A}

or

RPTOR1^{C269A} HA - DrugR    Cl2::pRib-RPTOR1^{C269A}-HA

pRib expression cassette

**Figure EV4. Secondary sequences alignments using Alphafold model of RPTOR1.**

(A) The X-ray crystal structures of HsCaspase-7 (PDB:1F1J), AtRAPTOR1 (PDB:5WBI), and the Alphafold model of LinfRPTOR1 (LINF_250011400) are shown individually on the left hand panels. The active site cysteine residues are denoted by the labels. The AlphaFold model (AF-A4I1A2-F1-model_v3.pdb) was downloaded from the AlphaFold Protein Structure Database (AlphafoldDB). AtRAPTOR1 and LinfRPTOR1 were superposed in UCSF Chimera using the MatchMaker tool (right hand panels). (B) The Match->Align tool was used to generate amino acid alignments from the structural superposition. Residues are coloured using the ClustalX scheme. The active site cysteines are annotated by stars above the sequence. (C) RPTOR1 complementation lines were generated by integrating untagged or HA-tagged WT and C269A *RPTOR1* into the ribosomal locus of *RPTOR1*<sup>-/flox</sup> Cl2 using homologous recombination.

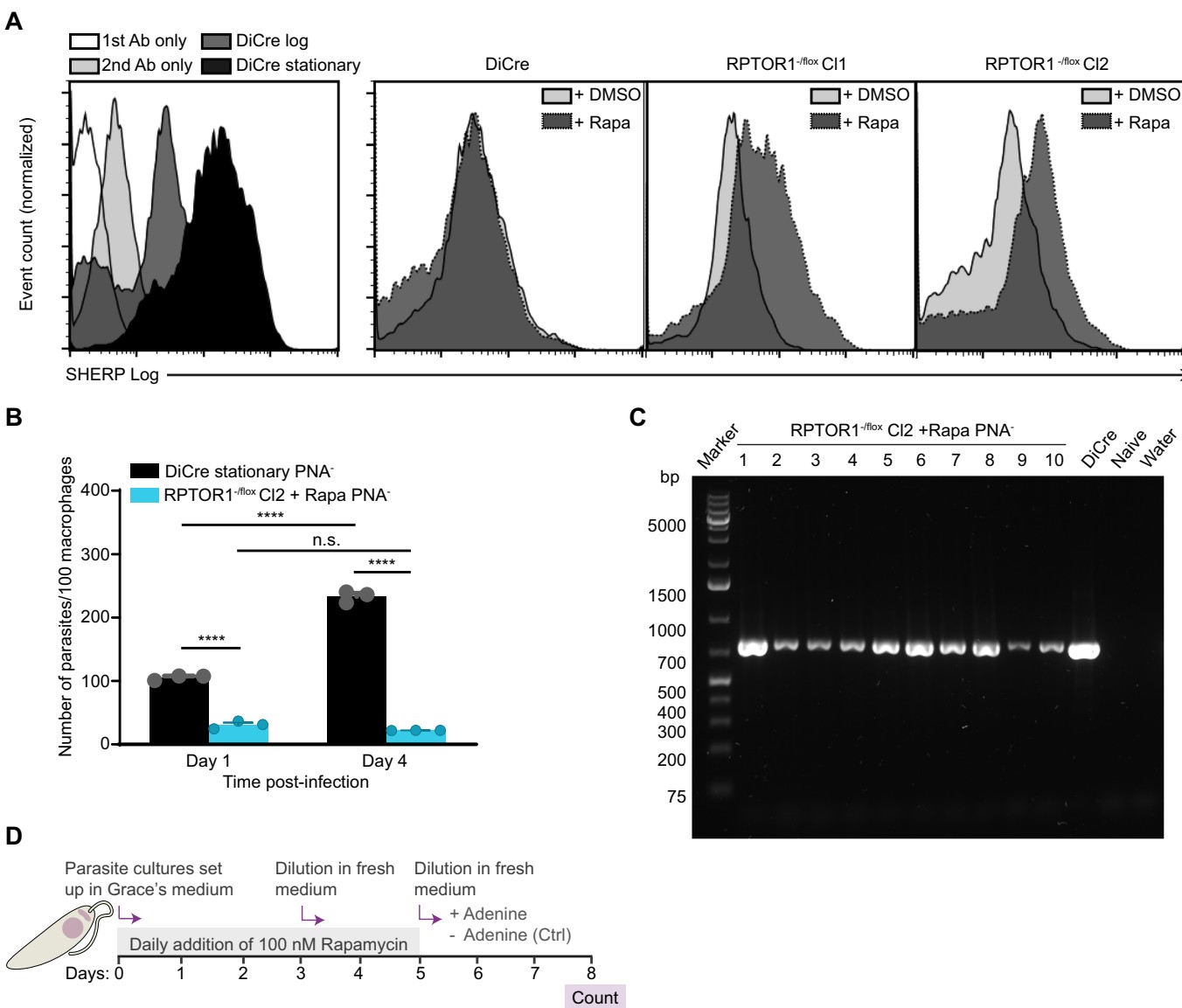

**Figure EV5. RPTOR1 loss induces metacyclogenesis but is detrimental for murine infection.**

(A) Flow cytometry analysis of SHERP expression. Log-stage promastigotes were treated for three days with DMSO or rapamycin (+Rapa), diluted and cultured for three more days with daily DMSO and rapamycin treatment. Cells were then fixed, permeabilized and stained with anti-SHERP and Alexa Fluor 647 (AF647)-conjugated secondary antibodies. Histograms of AF647 fluorescence (SHERP staining) in control (left panel) and DMSO or rapamycin treated cells. Controls (left panel) include DiCre cells stained with primary (1st Ab) or secondary antibody (2nd Ab) only and SHERP-stained DiCre early-log or stationary-phase cells. (B) Macrophage infectivity of PNA⁻ promastigotes. DiCre cells were cultured for 7 days in the presence of rapamycin to reach stationary phase. *RPTOR1⁻/flox* cells were induced with rapamycin for three days, diluted in fresh medium and cultured for three more days with daily addition of rapamycin. Metacyclic promastigotes (PNA⁻) were then purified from cultures by PNA agglutination and added to thioglycollate-elicited peritoneal macrophages at a 6:1 ratio (parasites:macrophages). Cells were analysed at day 1 and day 4 after infection using microscopy. Graph shows values of triplicate wells from one experiment as dots and their mean ± SD. A repeat experiment using an infection ratio of 3:1 is shown in Fig. 6A. ***$P$ value ≤ 0.001 in a two-way ANOVA with Bonferroni post hoc test. n.s. not significant. (C) PCR analysis of genomic DNA from ears of BALB/c mice that were infected with PNA⁻ promastigotes. Ears were harvested at 6 wks p.i., digested with collagenase D and the genomic DNA extracted. To assess the presence of escape mutants (with un-excised RPTOR1) PCR was performed using primers that amplify a 700 bp fragment from the RPTOR1 CDS. gDNA from the ears of ten mice infected with rapamycin-induced *RPTOR1⁻/flox*, one mouse infected with DiCre (positive control), one naïve mouse (negative control) or water (negative control) was used as template. Samples originated from the same ears represented in Fig. 6F and Fig. 6G. (D) Outline of adenine response experiment. DiCre, *RPTOR1⁻/flox* Cl2 and *RPTOR1* complementation lines were grown in Grace's medium with daily addition of 100 nM rapamycin or DMSO for five days. After three days of induction cells were diluted in fresh medium. On day 5 cells were diluted again with the addition or not of 500 µM adenine and counted after three days (day 8). Source data are available online for this figure.

