## [Peer Review File · EMBO Reports]

TORC1 is an essential regulator of nutrient-controlled proliferation and differentiation in *Leishmania*

Elmarie Myburgh, Vincent Geoghegan, Eliza Alves-Ferreira, Y. Romina Nievas, Jaspreet Grewal, Elaine Brown, Karen McLuskey, and Jeremy Mottram

Corresponding author(s): Elmarie Myburgh (ELMARIE.MYBURGH@YORK.AC.UK)

Review Timeline:

Submission Date:	16th Jan 23
Editorial Decision:	7th Mar 23
Revision Received:	29th Oct 23
Editorial Decision:	14th Nov 23
Revision Received:	19th Jan 24
Accepted:	25th Jan 24

Editor: Achim Breiling

Transaction Report:

Dear Dr. Myburgh,

Thank you for the transfer of your manuscript to EMBO reports. I have now received the reports from the three referees that were asked to evaluate your study, which can be found at the end of this message.

As you will see, the referees indicate that these findings are of interest. However, they have several comments, concerns, and suggestions, indicating that a major revision of the manuscript is necessary to allow publication of the study in EMBO reports. As the reports are below, and all the referee concerns need to be addressed as indicated in the reports, I will not detail them here.

Given the constructive referee comments, I would like to invite you to revise your manuscript with the understanding that all referee concerns must be addressed in the revised manuscript and in a detailed point-by-point response. Acceptance of your manuscript will depend on a positive outcome of a second round of review. It is EMBO reports policy to allow a single round of revision only and acceptance of the manuscript will therefore depend on the completeness of your responses included in the next, final version of the manuscript.

- 1) a .docx formatted version of the final manuscript text (including legends for main figures, EV figures and tables), but without the figures included. Figure legends should be compiled at the end of the manuscript text.
- 2) individual production quality figure files as .eps, .tif, .jpg (one file per figure), of main figures (up to 8) and EV figures. Please upload these as separate, individual files upon re-submission.

- 3) a complete author checklist, which you can download from our author guidelines (<https://www.embopress.org/page/journal/14693178/authorguide>). Please insert page numbers in the checklist to indicate where the requested information can be found in the manuscript. The completed author checklist will also be part of the RPF.

- 4) that primary datasets produced in this study (e.g. RNA-seq, ChIP-seq, structural and array data) are deposited in an appropriate public database. If no primary datasets have been deposited, please also state this in a dedicated section (e.g. 'No primary datasets have been generated and deposited'), see below.

The accession numbers and database should be listed in a formal "Data Availability" section (placed after Materials & Methods) that follows the model below. This is now mandatory (like the COI statement). Please note that the Data Availability Section is restricted to new primary data that are part of this study. This section is mandatory. As indicated above, if no primary datasets have been deposited, please state this in this section

Data availability

8) Regarding data quantification and statistics, please make sure that the number "n" for how many independent experiments were performed, their nature (biological versus technical replicates), the bars and error bars (e.g. SEM, SD) and the test used to calculate p-values is indicated in the respective figure legends (also for potential EV figures and all those in the final Appendix). Please also check that all the p-values are explained in the legend, and that these fit to those shown in the figure. Please provide statistical testing where applicable. Please avoid the phrase 'independent experiment', but clearly state if these were biological or technical replicates. Please also indicate (e.g. with n.s.) if testing was performed, but the differences are not significant. In case n=2, please show the data as separate datapoints without error bars and statistics. See also: <http://www.embopress.org/page/journal/14693178/authorguide#statisticalanalysis>

9) Please add scale bars of similar style and thickness to all the microscopic images, using clearly visible black or white bars (depending on the background). Please place these in the lower right corner of the images themselves. Please do not write on or near the bars in the image but define the size in the respective figure legend.

10) Please also note our reference format:

12) We now use CRediT to specify the contributions of each author in the journal submission system. CRediT replaces the author contribution section. Please use the free text box to provide more detailed descriptions and remove the author contributions from the manuscript. See also guide to authors:

<https://www.embopress.org/page/journal/14693178/authorguide#authorshipguidelines>

13) Please upload the information in the 'Reagents and Tools table' separately and remove this from the manuscript text. I have attached templates for that in word or excel format. Please upload the filled in table to the manuscript tracking system as a 'Reagent Table' file. The example below shows how the table will display in the published article and includes examples of the type of information that should be provided for the different categories of reagents and tools. Please list your reagents/tools

using the categories provided in the template and do not add additional subheadings to the table. Reagents/tools that do not fit in any of the specific categories can be listed under "Other":

https://www.embopress.org/pb%2Dassets/embo-site/msb_177951_sample_FINAL.pdf

14.) Please restrict the number of keywords to 5 and order the manuscript sections like this, using these names:

Title page - Abstract - Keywords - Introduction - Results - Discussion - Materials and Methods - Data availability section - Acknowledgements - Disclosure and Competing Interests Statement - References - Figure legends - Expanded View Figure legends

I look forward to seeing a revised version of your manuscript when it is ready. Please let me know if you have questions or comments regarding the revision.

Yours sincerely,

Referee #1:

The manuscript 'TORC1 is an essential regulator of nutrient-dependent differentiation in Leishmania' investigates interactions and functions of scaffolding protein RPTOR1 combining endogenous tagging, immuno-precipitation, MS analysis and gene ablation using a conditional, DiCre-based approach. Most of the manuscript is dedicated to an in-depth description of the null mutant phenotype with respect to in vitro promastigote proliferation, protein synthesis, morphology, metacyclogenesis and infection. The manuscript is well written, and the studies are of high quality and well controlled. The impact of this work is somewhat limited given its rather descriptive nature that does not further analyze how RPTOR1-dependent mTORC1 activities are controlling the various functions associated with the null mutant phenotype. The impact could have been considerably strengthened by assessing the down-stream targets of this complex and monitoring mTORC1 activity either directly (in vitro kinase assay) or indirectly (phosphorylation of substrates). Furthermore, some of the interpretations are not supported by the results as indicated in detail below.

Figure 1: The authors indicate that they use 'quantitative analysis of complexes by mass spectrometry' (line 99). The volcano plot shown in Fig. 1A gives the somewhat misleading impression that the shown biological interactions are statistically significant calculating the 'average probability of true interaction' based on spectral count. While this method may test significance in terms of MS analysis, it does not establish a statistical proof that these interactions are reproducible across independent experiments. Further validation is only provided for the interaction RPTOR1 and TOR1 (Fig. 1B) calling into question the other interactions, which were seen in only one experiment. Even though these putative interactions are not further pursued, the data are shown again in Fig. 1B and these non-validated interactions are discussed in some depth in the Discussion section. Given the lack of validation of these interactions, these parts should be toned down (or their specific interaction validated).

Figures 2 and 3: The authors mention in the text the generation of heterozygous mutants (line 133). Was there any phenotype observed for those? The authors convincingly show that deletion of the target gene causes cell cycle arrest without affecting cell viability. This phenotype is observed in response to various forms of stress and has been linked to reduced protein synthesis due to eIF2-alpha phosphorylation. Indeed, reduced protein synthesis is shown in Fig. 3B. Here the question arises if the observed phenotypic changes are directly regulated by RPTOR1/mTORC1, or if they are an indirect due to endogenous stress caused by the deletion. A direct assessment of mTORC1 activity could provide data to better distinguish between these possibilities and a direct link the mutant phenotype to mTORC1 signaling would considerably increase the impact of this manuscript.

Figure 4: In my opinion this figure (and most of the results shown) is obsolete as reasoning is based on rather weak structural models and sequence alignments. Indeed, the results are negative, and the initial (weak) hypothesis disproven. The only results to keep here are those showing that the RPTOR1 add back line regains WT phenotype.

Figure 5: The morphological analysis shown in Fig. 3 would much better fit here. The finding that RPTOR1 deletion induces bona fide metacyclic parasites (based on the agglutination tests) is the strongest piece of information. The authors speculate that increased expression of RPTOR1 may be associated with reduced metacyclogenesis (line 298). This should be followed up by episomal over-expression of a non-tagged version of RPTOR1. They then study the capacity of these mutants to establish infection in macrophages and mice. They use the term 'infectivity' as synonymous to uptake. Infectivity is defined as the capacity to establish productive infection. In Fig. 5D, they simply follow the level of parasite uptake by macrophages, which is a passive process. Thus, 'infectivity' should be replaced by 'efficiency to be ingested by macrophages' or similar. The authors speculate that the null mutant parasites 'may differentiate into amastigotes' (line 317). I am surprised they do not provide a simple IFA

(including a nuclear stain) to prove this point. These data need to be added. Furthermore, the team has previously applied their conditional ablation system on intracellular amastigotes. The manuscript would significantly gain in impact assessing the essential nature of RPTOR1 on the amastigote stage, which would also elucidate the mechanism of virulence attenuation (bad metacyclics or bad amastigotes).

Figure 6: The model is confusing and could be visually more appealing, for example using the biorender package.

Discussion: The discussion is way too long (over 4 pages), diluted with too much background information, and shows substantial redundancy to the results section, which is often symptomatic for descriptive papers that lack mechanistic insight. Also, there is too much discussion on some of the non-validated interactions.

Referee #2:

This study defines the composition of the TORC1 signalling complex in *Leishmania* and the potential roles of one of the core components, RPTOR, in growth control, nutrient sensing and infectivity. While previous studies have suggested that TOR1 is essential in these parasites, dissection of the roles of this universal eukaryotic signalling complex in these divergent protists has been hampered by the inability to generate knock-out lines. In this study, the authors utilize proteomics/pull down approaches to show that the putative *Leishmania* TOR1 homologue is part of a complex that contains RPTOR1 and other proteins that are both unique and conserved in other eukaryotic TORC1 complexes. Conditional disruption of RPTOR1 using the newly developed diCre system (and by inference the TORC1 complex) leads to promastigote G1 cell cycle growth arrest, and premature differentiation to non-dividing metacyclic promastigotes. Parasite infectivity in a murine infection model is also effectively inhibited. Finally, evidence is provided indicating that the caspase domain of *Leishmania* RPTOR1 lacks protease activity (in common with some other eukaryotes) and that TORC1 is required for parasite response to purine availability. Overall, this work provides new insights into the role of RPTOR1 and the TORC1 complex in regulating *Leishmania* growth and virulence. While the reported phenotyping analyses have been carefully and thoroughly done, further characterization of virulence phenotype in macrophages and mice would add to the study and is needed to support some of the conclusions. Additional information on the subcellular localization of the complex would also help to define how the parasite complex contributes to nutrient sensing.

Major points to address

The authors undertake experiments in both *L. major* (diCre KO) and *L. mexicana* (complex analysis). Although the species used are indicated in the Experimental section, the use of different species is not explicitly indicated in the results and discussion sections. Given that several aspects of the biology of each species differ (including metacyclogenesis), it is important to note which species is being used in each experiment throughout the text and figure legends.

The complete loss of parasite infectivity of the RPTOR1^{-/-} mutant in the BALB/c ear infection model is striking, particularly given that growth of the mutant in BMM was apparently unaffected. However, further characterization of both the macrophage and BALB/c infection phenotypes is warranted. For example, have the authors shown that promastigotes differentiate to amastigotes in infected BMM and if so, is there any difference in the rate at which this occurs between control and RPTOR1^{-/-} parasites? Similarly, while measurement of lesion size in the BALB/c mice infections demonstrates that virulence of the RPTOR1^{-/-} mutant is attenuated in the mammalian host, it remains possible that mutant parasites are able to establish long term, asymptomatic infections. Assessment of tissues from the site of infection and/or local lymph nodes for quantitation of viable parasites is required before concluding whether metacyclic promastigotes are unable to establish infection in animals and the fate of any amastigotes generated.

In other eukaryotes, TORC1 complexes are assembled on the lysosome or other organelles in the secretory pathway. The authors have expressed GFP and epitope tagged versions of RPTOR1 and it should be relatively straightforward to determine whether the complexes are associated with organellar fractions by fluorescence microscopy and/or subcellular fractionation. This information would significantly add to the study and general conclusions as to the extent that TORC1 signalling is conserved in protists.

Minor comments

What is known about the control bait Ser/Thr protein kinase (Imx.29.3580) used in the pull-down experiments and why specifically was it chosen?

Referee #3:

This is an important paper on the TORC1 complex in *Leishmania*, using a variety of up to date methodologies. The authors show that the parasite TORC1 interacts with RPTOR1 and another protein, LST8. There are four TORC complexes in *Leishmania*, which vary in their physiological effects. The authors use a variety of cutting-edge methodologies, to investigate the function of TORC1 and its binding partner RPTOR1. Since it was not possible to delete TORC1, possibly indicating that it is

essential, the authors attempted to manipulate expression of RPTOR1 by deleting a single allele using CRISPR-Cas9. In addition they generated a conditional deletion mutant of RPTOR1 using the DiCre methodology, followed by deletion of the second allele using homologous recombination. Having these mutants, the authors showed that loss of RPTOR1 resulted in cell-cycle arrest of the cells in the G1 phase of the cell cycle. They also showed that RPTOR1 controls protein synthesis and cell size. RPTOR1 loss induced parasite differentiation to metacyclic-like promastigotes, including their surface antigens. However, it was not subject to regulation by nutrient availability. The authors suggest that TORC1 functions downstream to the parasite mechanism that senses purine availability.

Using a bioinformatics approach the authors showed that RPTOR1 contains a predicted caspase domain. They further showed that recovery of RPTOR1 expression by reintroducing a transgenic RPTOR1 recovered the phenotype of the floxed RPTOR1. However, disruption of the potential and predicted catalytic site cysteine in the caspase-like domain had no effect on the complemented function of RPTOR1. The authors suggest that RPTOR1 caspase activity is not required for promastigote proliferation, although there is no evidence that this domain is functional in the parasite RPTOR1.

Points to be addressed prior to publication:

- 1) Figures 4 and EV5 - The authors should provide details on how they generated the alpha fold predicted structure. Was the structure generated through the on-line server through UniProt or did they perform the prediction with downloaded programs?
- 2) The authors should provide technical details on how the scheme in Figure 4B was generated. This should also be described in the Materials and Methods section. Furthermore, the structure prediction could be further verified by using another program, such as the ESM fold.
- 3) Figure EV1 B and D - the authors should provide further details on how their gels were loaded - what do they mean when they say that "a sample" was taken from the lysate, of the pulled down fractions? Do the samples represent the same number of cells in the different cell lines?
- 4) Figure EV6 - how did the authors generate the sequence alignment that was based on the secondary structures? What program did they use? What do they mean by indicating that there were missing residues from the caspase-7 structure (cleaved loop) in the Arabidopsis AtRPTOR1? Were they actually missing, or did they fail to align? Are these residues missing also from the Leishmania protein? Could these residues account for the missing proteolytic activity in the parasite RPTOR1?
- 5) Text: Page 10 The authors should rephrase their conclusions on the requirement for the catalytic function of the assumed proteolytic activity of RPTOR. Although no proteolytic activity was shown, the authors concluded that such predicted activity was not required for promastigote proliferation and growth, since replacement of the missing RPTOR1 with a protein that was mutated in its cys269 residue enabled the phenotypic recovery of the cells. Do the authors have any evidence that the caspase-7 domain is functional in the Leishmania RPTOR1? Can the authors provide a predicted function for the proteolytic activity of the protein? Is there any other evidence for such an activity other than the structural homologies, that may be misleading?
- 6) The title: The authors write that nutritional depletion leads to "premature differentiation from proliferative promastigotes to non-dividing mammalian-infective metacyclic forms. These parasites cannot develop into proliferative amastigotes in the mammalian host, or respond to nutrients to differentiate to proliferative retroleptomonads, which are required for their blood-meal induced amplification in sand flies and enhanced mammalian infectivity". In such case the title that declares that TORC1 is an essential regulator of nutrient-dependent differentiation is not fully accurate, since the differentiation process varies from that observed in reality. I would therefore rephrase the title more carefully.

EMBOR-2023-56840V2: Response to reviewers

We thank the reviewers for their thorough evaluation of the manuscript and for their supportive comments and constructive feedback. In the revised manuscript we have included new experimental data to address the points raised by the referees. These additional/revised data have been incorporated in the Main Figures, Expanded View figures and Appendix under the headings below. We have also included changes according to editorial requests and the author checklist.

The points made by the three referees are addressed in detail following our list of additional/revised figures and "other changes to the manuscript". Our responses are in blue. Changes in the attached manuscript and supplementary files are also highlighted.

Additional/revised data

Figure 1. Characterization of the RPTOR1 containing complex in *Leishmania*

- **New Figure 1D:** Localisation of RPTOR1 using live cell fluorescence microscopy. (shows localisation of RPTOR1 in endomembranes and cytosol)

Figure 3. RPTOR1 controls protein synthesis and cell size.

- **Revised:** Panel E and F moved to Figure 5 (suggested by reviewer 1)

Figure 5. RPTOR1 loss induces metacyclogenesis.

- **Revised figure 5** that contains panel A & B (from original Figure 3E & F) with C to E (from original Figure 5 A-C).

Figure 6. RPTOR1^{-/-} parasites can differentiate to amastigotes but are unable to proliferate in macrophages or mice.

- **New Figure 6A:** Macrophage infectivity of PNA⁻ promastigotes. (shows lack of proliferation by RPTOR1^{-/-} parasites in peritoneal macrophages between 3 hrs and day 4 post-infection)
- **New Figure 6B:** Amastigotes isolated from infected peritoneal macrophages. (shows amastigotes isolated from infected peritoneal macrophages)
- **New Figure 6C:** Aspect ratio (body length relative to body width) of purified metacyclic promastigotes before infection (0 hr) and amastigotes isolated at day 5 after infection. (shows amastigotes isolated from infected peritoneal macrophages).
- **New Figure 6D & E:** EdU incorporation in amastigotes within thioglycollate-elicited peritoneal macrophages at day 5 after infection. (new data showing that RPTOR1^{-/-} parasites do not proliferate as amastigotes in peritoneal macrophages)
- **New Figure 6D & E:** EdU incorporation in amastigotes within thioglycollate-elicited peritoneal macrophages at day 5 after infection. (shows that RPTOR1^{-/-} parasites do not proliferate as amastigotes in peritoneal macrophages)
- **New Figure 6G:** Parasite load in ears of infected mice determined by qPCR. (shows relative quantification of parasites in the ears of infected mice confirming lower parasite loads of RPTOR1^{-/-} compared to DiCre)
- **Revised Figure 6F, H and I** (moved here from original Figure 5E, F and G)

Figure EV3. RPTOR1 is essential for cell proliferation and long-term survival.

- **Revised Figure EV3B:** Cell densities of uninduced (+DMSO, solid line) and rapamycin-induced (+Rapa, dashed line) cells. (amended to include a RPTOR1 heterozygote line and showing no proliferation defect in this line)

Figure EV4 Secondary sequences alignments using Alphafold model of RPTOR1.

- Original Figure EV4 moved to new Appendix S1 to limit Expanded View figures to five.
- Original Figure EV5 is now Figure EV4

Figure EV5 (previously EV6). RPTOR1 loss induces metacyclogenesis but is detrimental for murine infection.

- **Original Figure EV6A** has been moved to **Appendix S2** to accommodate new Figure EV6B and C; original panel B has been moved to A.
- **New Figure EV5B:** Macrophage infectivity of PNA⁻ promastigotes. (shows lack of proliferation by *RPTOR1*^{-/-} parasites in peritoneal macrophages between day 1 and day 5 post-infection - a repeat experiment to support new Figure 6A)
- **New Figure EV5C:** PCR analysis of genomic DNA from ears of BALB/c mice that were infected with PNA⁻ promastigotes. (shows that all rapamycin-induced RPTOR1^{-/flox} infected ears are positive for RPTOR1 CDS)
- Figure title amended to include description of above data and original panel C has been moved to D.

Figure 7. A proposed model of *Leishmania* proliferation or differentiation in response to nutrients and active TORC1.

- Revised model due to the additional data showing that *RPTOR1*^{-/-} differentiate to amastigotes but do not proliferate in macrophages and mice.

Appendix Figure S1 and S2

- Figure EV4 has been moved to appendix S1 to limit the number of EV figures to five; Figure EV6 is now Figure EV5.
- Figure EV6A has been moved to appendix S2 to accommodate new panel B (repeat infected peritoneal macrophage data to support Figure 6A) and panel C (endpoint PCR data for ear samples from mice).

Other changes to the manuscript (editorial requests/author checklist)

1. We have added details of animal experiments into Materials and Methods which includes mouse strain, sex, age, sample size, randomization, blinding and sample size calculations.
2. We have added individual points to graphs and updated statistics and description of replicates according to editorial guidelines.
3. We have reduced Expanded View figures to 5 (other figures have been moved into the appendix as described above).
4. A separate Reagents and Tools table has been submitted.

Point-by-point Responses to reviewers

Referee #1:

The manuscript 'TORC1 is an essential regulator of nutrient-dependent differentiation in Leishmania' investigates interactions and functions of scaffolding protein RPTOR1 combining endogenous tagging, immuno-precipitation, MS analysis and gene ablation using a conditional, DiCre-based approach. Most of the manuscript is dedicated to an in-depth description of the null mutant phenotype with respect to in vitro promastigote proliferation, protein synthesis, morphology, metacyclogenesis and infection. The manuscript is well written, and the studies are of high quality and well controlled. The impact of this work is somewhat limited given its rather descriptive nature that does not further analyze how RPTOR1-dependent mTORC1 activities are controlling the various functions associated with the null mutant phenotype. **The impact could have been considerably strengthened by assessing the down-stream targets of this complex and monitoring mTORC1 activity either directly (in vitro kinase assay) or indirectly (phosphorylation of substrates).** Furthermore, some of the interpretations are not supported by the results as indicated in detail below.

1. Figure 1: The authors indicate that they use 'quantitative analysis of complexes by mass spectrometry' (line 99). The volcano plot shown in Fig. 1A gives the somewhat misleading impression that the shown biological interactions are statistically significant, calculating the 'average probability of true interaction' based on spectral count. While this method may test significance in terms of MS analysis, it does not establish a statistical proof that these interactions are reproducible across independent experiments. Further validation is only provided for the interaction RPTOR1 and TOR1 (Fig. 1B) calling into question the other interactions, which were seen in only one experiment. Even though these putative interactions are not further pursued, the data are shown again in Fig. 1B and these non-validated interactions are discussed in some depth in the Discussion section. Given the lack of validation of these interactions, these parts should be toned down (or their specific interaction validated).

Figure 1A are plots of fold enrichment against probability for each identified protein. This is a standard way of presenting affinity purification mass spectrometry data and we disagree that these plots are a misleading representation of the data. In the corresponding figure legend we describe SAINT scores as 'average probability of true interaction', terminology that the original authors of the SAINT software use in their publication (PMID: 21131968). However, we agree that this particular terminology is ambiguous and have simplified it to describing SAINT scores as 'average probability across three biological replicates'. It is important to note that 4 of the 7 proteins that co-purify with RPTOR1/TOR1 are homologues of known components of mammalian TORC1 and HIP likely co-purifies with HSP83-1. The remaining 2 are uncharacterised proteins which co-purify when 2 different baits are used to capture the complex, increasing the confidence that these are true interactors. We feel that the AP-MS strategy used was stringent (endogenous tagging with small tag, gentle elution from beads with biotin) and appropriately controlled (comparison with an unrelated affinity purified cytoplasmic protein). This is reflected in the small, distinct subset of proteins we detect as significantly co-

purifying with TOR1/RPTOR1. Follow up of all detected interactors is not the focus of the presented manuscript therefore, as suggested, we have reduced the depth of discussion dedicated to these. Nevertheless, we believe it is important to clearly present the results of our AP-MS experiment as they could be used for hypothesis generation by other groups working in the field or other researchers who become interested in any of these proteins via other routes.

2. Figures 2 and 3: The authors mention in the text the generation of heterozygous mutants (line 133). Was there any phenotype observed for those?

We generated heterozygotes containing one WT allele (*RPTOR*^{+/-}); these parasites had similar promastigote proliferation compared to DiCre and were not analysed for other phenotypes. These data have now been added to the manuscript (new Figure EV3B and lines 170-172). Inducible heterozygotes (*RPTOR1*^{+/*flox*}) were not analysed for the phenotypes described in our manuscript.

The authors convincingly show that deletion of the target gene causes cell cycle arrest without affecting cell viability. This phenotype is observed in response to various forms of stress and has been linked to reduced protein synthesis due to eIF2-alpha phosphorylation. Indeed, reduced protein synthesis is shown in Fig. 3B. Here the question arises if the observed phenotypic changes are directly regulated by RPTOR1/mTORC1, or if they are an indirect due to endogenous stress caused by the deletion. A direct assessment of mTORC1 activity could provide data to better distinguish between these possibilities and a direct link the mutant phenotype to mTORC1 signaling would considerably increase the impact of this manuscript.

There is sufficient supporting evidence in this paper and from published literature that the phenotypes observed here directly relate to the *RPTOR1* gene knockout rather than the stress caused by the deletion of any gene: 1) deletion of other non-essential genes do not result in the growth arrest observed here e.g. Damanesco et al (2020, PLOS genetics 16(7), DOI: [10.1371/journal.pgen.1008828](https://doi.org/10.1371/journal.pgen.1008828)) deleted *PIF6* in *L. major* using the inducible DiCre system and observed no defects in proliferation or DNA synthesis in the time scales used for our experiments. 2) Our complementation mutant lines were generated using the inducible KO line thus rapamycin induction of these lines also cause deletion of the floxed gene as it does for the inducible KO line. As shown in Figure 4E, F and G these complementation lines do not have defects in proliferation, growth and cell cycle progression after rapamycin induction. The additional *RPTOR1* gene in the ribosomal RNA locus rescues these parasites from the phenotypes observed for the knockout, indicating that the phenotype is dependent on the complete deletion of RPTOR1. This is pointed out in the results section (lines 257-260).

Regarding the last point, we agree that directly assessing TORC1 activity would be ideal but since the targets for TORC1 in *Leishmania* have not been defined this is not easily done at the current time. Lines 428-436 in the Discussion point out that orthologues for the mTORC1 substrates, p70 S6 kinase and 4E-BP1 that are used to assess mTORC1 activity in other systems have not been identified in trypanosomatids nor have other substrates been described.

3. Figure 4: In my opinion this figure (and most of the results shown) is obsolete as reasoning is based on rather weak structural models and sequence alignments. Indeed, the results are negative, and the initial (weak) hypothesis disproven. The only results to keep here are those showing that the RPTOR1 add back line regains WT phenotype.

We would like to keep these results in the manuscript. Structural models and sequence alignments are often the starting point for defining functions of unknown genes especially for less studied organisms such as *Leishmania*, where many gene functions are still unknown. Our results show that the cysteine is not required for the assessed functions of RPTOR1, but this should not exclude them from publication. Negative results remain an important part of the scientific process and discovery.

4. Figure 5: The morphological analysis shown in Fig. 3 would much better fit here.

We have amended our figures and ms text - the morphological analysis of promastigotes (originally in Figure 3E and F has now been moved to new Figure 5A and B). To make space for this in Figure 5 we have made a new Figure 6 with additional amastigote data to address the later reviewer comments.

The finding that RPTOR1 deletion induces bona fide metacyclic parasites (based on the agglutination tests) is the strongest piece of information. The authors speculate that increased expression of RPTOR1 may be associated with reduced metacyclogenesis (line 298). This should be followed up by episomal over-expression of a non-tagged version of RPTOR1.

We have removed this speculation since we do not follow this up with additional data and evidence of overexpression.

5. They then study the capacity of these mutants to establish infection in macrophages and mice. They use the term 'infectivity' as synonymous to uptake. Infectivity is defined as the capacity to establish productive infection. In Fig. 5D, they simply follow the level of parasite uptake by macrophages, which is a passive process. Thus, 'infectivity' should be replaced by 'efficiency to be ingested by macrophages' or similar.

We showed infection data at day 5 post-infection (p.i.). At this late stage the presence of parasites in macrophages indicate infection and differentiation to amastigotes, not just uptake. Parasites that are non-infective will not survive inside macrophages for this period even if taken up by passive processes. Uptake is usually assessed in early time points (2-6 hrs p.i) but parasites that are not infective will decline after this as they cannot survive in macrophages without transforming to amastigotes from metacyclic promastigotes (e.g. *CPB* null mutants in Casgrain et al, *PlosPathogens* 12(5), 2016, <https://doi.org/10.1371/journal.ppat.1005658> and *spt2* null mutants in Zhang et al, *Mol Micro*, 2005, <https://doi.org/10.1111/j.1365-2958.2005.04493.x>).

To distinguish between defects in uptake, amastigogenesis or amastigote proliferation we have now added additional macrophage data (new Figure 6A-E). This shows that

the *RPTOR1*^{-/-} mutants are taken up by macrophage in similar numbers to DiCre (assessed at 3 hrs post-infection) and are still present by day 4 and 5 post-infection but do not proliferate as amastigotes after uptake. We have also modified the manuscript text to specify uptake vs proliferation.

6. The authors speculate that the null mutant parasites 'may differentiate into amastigotes' (line 317). I am surprised they do not provide a simple IFA (including a nuclear stain) to prove this point. These data need to be added.

We agree that confirming that the null mutant parasites differentiate to amastigotes is important. We have addressed this by isolating the parasites from 5-day infected peritoneal macrophages, staining parasites using anti-*Leishmania* OPB antibodies and DAPI (to stain DNA) and imaging them by fluorescence microscopy. This confirmed that they are amastigotes based on morphology. New Figure 6B and C, results section in line 319-336 and discussion (lines 471-475) have been added.

7. Furthermore, the team has previously applied their conditional ablation system on intracellular amastigotes. The manuscript would significantly gain in impact assessing the essential nature of RPTOR1 on the amastigote stage, which would also elucidate the mechanism of virulence attenuation (bad metacyclics or bad amastigotes).

It has been problematic to analyse amastigotes using this conditional ablation system. While genes can be excised in amastigotes using the system (as shown in Duncan et al, 2016) amastigotes are sensitive to rapamycin and their further growth are not comparable to WT parasites independent of the gene being excised. This is a deficiency of the diCre system, and this sensitivity is highlighted in Duncan et al, 2016, Mol Micro 100(6) (DOI: [10.1111/mmi.13375](https://doi.org/10.1111/mmi.13375)) and a later review (Duncan et al, 2017, Mol Biochem Parasitol 216: , [doi: 10.1016/j.molbiopara.2017.06.005](https://doi.org/10.1016/j.molbiopara.2017.06.005)). Due to this sensitivity parasites are induced for gene excision as promastigotes and differentiated to amastigotes to assess the effect on their growth in macrophages and mice. Unfortunately, this doesn't allow the distinction between a defect in differentiation from promastigotes to amastigotes or a defect in amastigote proliferation. This is complicated further by the use of *L. major* parasites in our study – the strain of parasites we use cannot be grown as axenic amastigotes.

In order to analyse this we have done more in-depth analysis of our null mutant parasites in peritoneal macrophages. Our previous experiments were performed in bone-marrow derived macrophages and a paper from Mandell et al (Mandell *et al*, 2022, PLOS Negl Trop Dis 16(10), DOI: [10.1371/journal.pntd.0010893](https://doi.org/10.1371/journal.pntd.0010893)) showed that *L. major* amastigotes do not proliferate in this type of macrophage in the 3 days that they analysed them. On the other hand, amastigotes could proliferate in peritoneal macrophages in their study. Our new data in peritoneal macrophages show that 1) *RPTOR1*^{-/-} promastigotes differentiate to amastigotes (discussed in point 6 above) and 2) *RPTOR1*^{-/-} amastigotes do not proliferate (in contrast to the control DiCre amastigotes). This is shown through counting of intracellular parasites inside macrophages over time (3 hr, day1 and day 4 post-infection) and measuring EdU incorporation into parasites between day 1 and day 5 post-infection.

Additional data has been added in Figure 6B to E, Figure EV5B and text updated in lines 319-346 in results and lines 471-475 in discussion.

8. Figure 6: The model is confusing and could be visually more appealing, for example using the biorender package.

We have edited the model (new Figure 7) to incorporate changes based on our additional data and hope it is now clearer.

9. Discussion: The discussion is way too long (over 4 pages), diluted with too much background information, and shows substantial redundancy to the results section, which is often symptomatic for descriptive papers that lack mechanistic insight. Also, there is too much discussion on some of the non-validated interactions.

We have edited the discussion and removed more than 25%.

Referee #2:

This study defines the composition of the TORC1 signalling complex in *Leishmania* and the potential roles of one of the core components, RPTOR, in growth control, nutrient sensing and infectivity. While previous studies have suggested that TOR1 is essential in these parasites, dissection of the roles of this universal eukaryotic signalling complex in these divergent protists has been hampered by the inability to generate knock-out lines. In this study, the authors utilize proteomics/pull down approaches to show that the putative *Leishmania* TOR1 homologue is part of a complex that contains RPTOR1 and other proteins that are both unique and conserved in other eukaryotic TORC1 complexes. Conditional disruption of RPTOR1 using the newly developed diCre system (and by inference the TORC1 complex) leads to promastigote G1 cell cycle growth arrest, and premature differentiation to non-dividing metacyclic promastigotes. Parasite infectivity in a murine infection model is also effectively inhibited. Finally, evidence is provided indicating that the caspase domain of *Leishmania* RPTOR1 lacks protease activity (in common with some other eukaryotes) and that TORC1 is required for parasite response to purine availability. Overall, this work provides new insights into the role of RPTOR1 and the TORC1 complex in regulating *Leishmania* growth and virulence. While the reported phenotyping analyses have been carefully and thoroughly done, **further characterization of virulence phenotype in macrophages and mice** would add to the study and is needed to support some of the conclusions. Additional information on the **subcellular localization** of the complex would also help to define how the parasite complex contributes to nutrient sensing.

Major points to address

10. The authors undertake experiments in both *L. major* (diCre KO) and *L. mexicana* (complex analysis). Although the species used are indicated in the Experimental section, the use of different species is not explicitly indicated in the results and discussion sections. Given that several aspects of the biology of each species differ (including metacyclogenesis), it is important to note which species is being used in each experiment throughout the text and figure legends.

We have edited the manuscript text including figure legends to clearly indicate when *L. major* or *L. mexicana* is used.

11. The complete loss of parasite infectivity of the *RPTOR1*^{-/-} mutant in the BALB/c ear infection model is striking, particularly given that growth of the mutant in BMM was apparently unaffected. **However, further characterization of both the macrophage and BALB/c infection phenotypes is warranted.** For example, have the authors shown that promastigotes differentiate to amastigotes in infected BMM and if so, is there any difference in the rate at which this occurs between control and *RPTOR1*^{-/-} parasites?

We have addressed this by more in-depth analysis of the parasites in macrophages. Mandell et al (2022, PLOS Negl Trop Dis 16(10), DOI: [10.1371/journal.pntd.0010893](https://doi.org/10.1371/journal.pntd.0010893)) reported that *L. major* parasites did not proliferate in BMMs but could do so in peritoneal macrophages. This may explain the lack of amastigote proliferation that we observed for our control DiCre lines in BMMs. We have now repeated this analysis using peritoneal macrophages to assess if there is a defect in differentiation to amastigotes and/or amastigotes proliferation in our null mutant line. Our new data show that *RPTOR1*^{-/-} metacyclic promastigotes differentiate to amastigotes but do proliferate in macrophages. This is also discussed in response to points 6 and 7 by reviewer 1. New Figure 6A to E, Figure EV5B and text in lines 319-346 in results and lines 471-475 in discussion have been added.

Assessing the rate of proliferation is more difficult as we cannot culture axenic amastigotes of *L. major*. The proliferation phenotype of *RPTOR1*^{-/-} mutant promastigotes results in low numbers that are available for macrophage infections making large scale purification of amastigotes for analysis difficult.

12. Similarly, while measurement of lesion size in the BALB/c mice infections demonstrates that virulence of the *RPTOR1*^{-/-} mutant is attenuated in the mammalian host, it remains possible that mutant parasites are able to establish long term, asymptomatic infections. Assessment of tissues from the site of infection and/or local lymph nodes for quantitation of viable parasites is required before concluding whether metacyclic promastigotes are unable to establish infection in animals and the fate of any amastigotes generated.

To address this issue, we've performed qPCR on the ear samples from the mice shown in Fig 5E of the original manuscript and now show relative quantification of the parasites present in the infected ears at the end-point of the experiment (6 wks post-infection). We also included an ear sample spiked with 1×10^5 DiCre parasites as a standard to give an estimate of parasite load in our samples. New Figure 6G has been added and the lesion size data has been moved to Figure 6F to accommodate additional figures as discussed in other points above. These new data show ~200-fold lower parasite load in ears of mice infected with the *RPTOR1*^{-/-} compared to DiCre metacyclic promastigotes. The parasite load in the *RPTOR1*^{-/-} infected mice is ~ 100-fold higher than in a naive ear suggesting that parasite DNA is still present in the mouse ears at a low level. The relative quantification to our standard suggests presence of less than 6×10^4 parasites in the ears of 9 mice with 1 ear showing a slightly higher load of $\sim 2.7 \times 10^5$ parasites. Possible explanations of this is that the *RPTOR1*^{-/-} parasites persist as non-proliferative amastigotes (as shown in macrophages) and that a small number of parasites with non-excised *RPTOR1* (excision escapees) persist and proliferate in mouse ears. We tried to perform qPCR to quantify excised vs non-excised *RPTOR1* parasites but were unsuccessful due to the low sensitivity of this approach. Instead, we carried out end-

point PCR on the gDNA samples from infected ears and show that the *RPTOR1* CDS is still present in gDNA from the ears of all ten mice that were infected with rapamycin-induced *RPTOR1*^{/flox} parasites. This suggests the presence of escape mutant parasites with non-excised *RPTOR1*. New Figure EV5C has been added to show these PCR results.

13. In other eukaryotes, TORC1 complexes are assembled on the lysosome or other organelles in the secretory pathway. The authors have expressed GFP and epitope tagged versions of RPTOR1 and it should be relatively straightforward to determine whether the complexes are associated with organellar fractions by fluorescence microscopy and/or subcellular fractionation. This information would significantly add to the study and general conclusions as to the extent that TORC1 signalling is conserved in protists.

To address this, we've stained a mNeonGreen-tagged RPTOR1 *L.mexicana* line with lysotracker and imaged them live using fluorescence microscopy to show where RPTOR1 is localised. Our data indicates some variability in RPTOR1 localization depending on the individual parasites imaged. In some parasites RPTOR1 co-localizes with lysotracker but in other parasites this is not the case making it difficult to conclusively say where RPTOR1 is localised. We have included these data in new Figure 1D and added text in lines 127-133 (results) and 435-437 (discussion). Materials and Methods have also been updated (lines 566-571 and 720-734).

Minor comments

What is known about the control bait Ser/Thr protein kinase (Lmx.29.3580) used in the pull-down experiments and why specifically was it chosen?

We selected Lmx.29.3580 as a control bait as it has a cytoplasmic localisation (Baker et al. 2021) and provides a more stringent control compared to a blank affinity purification. We have added a line (556-558) about why this control was chosen in the manuscript text.

Referee #3:

This is an important paper on the TORC1 complex in Leishmania, using a variety of up to date methodologies. The authors show that the parasite TORC1 interacts with RPTOR1 and another protein, LST8. There are four TORC complexes in Leishmania, which vary in their physiological effects. The authors use a variety of cutting-edge methodologies, to investigate the function of TORC1 and its binding partner RPTOR1. Since it was not possible to delete TORC1, possibly indicating that it is essential, the authors attempted to manipulate expression of RPTOR1 by deleting a single allele using CRISPR-Cas9. In addition they generated a conditional deletion mutant of RPTOR1 using the DiCre methodology, followed by deletion of the second allele using homologous recombination. Having these mutants, the authors showed that loss of RPTOR1 resulted in cell-cycle arrest of the cells in the G1 phase of the cell cycle. They also showed that RPTOR1 controls protein synthesis and cell size. RPTOR1 loss induced parasite differentiation to metacyclic-like promastigotes, including their surface antigens. However, it was not subject to regulation by nutrient

availability. The authors suggest that TORC1 functions downstream to the parasite mechanism that senses purine availability.

Using a bioinformatics approach the authors showed that RPTOR1 contains a predicted caspase domain. They further showed that recovery of RPTOR1 expression by reintroducing a transgenic RPTOR1 recovered the phenotype of the floxed RPTOR1. However, disruption of the potential and predicted catalytic site cysteine in the caspase-like domain had no effect on the complemented function of RPTOR1. The authors suggest that RPTOR1 caspase activity is not required for promastigote proliferation, although there is no evidence that this domain is functional in the parasite RPTOR1.

Points to be addressed prior to publication:

1) Figures 4 and EV5 - The authors should provide details on how they generated the alpha fold predicted structure. Was the structure generated through the on-line server through UniProt or did they perform the prediction with downloaded programs?

The AlphaFold model in EV5 (new EV4) was AF-A4I1A2-F1-model_v3.pdb that was downloaded from the AlphaFold Protein Structure Database (AlphafoldDB). Text to clarify this has now been added to the figure legend. Amendments to Figure 4 are discussed in responses below.

2) The authors should provide technical details on how the scheme in Figure 4B was generated. This should also be described in the Materials and Methods section. Furthermore, the structure prediction could be further verified by using another program, such as the ESM fold.

This information is already listed in the Methods and Materials section but more information has been added to clarify how the figure was generated (lines 803-807). The structures in Figure 4B are not structure predictions but topology diagrams based on known structures. The text in the legend (lines 1313-1314) has been changed to clarify this.

3) Figure EV1 B and D - the authors should provide further details on how their gels were loaded - what do they mean when they say that "a sample" was taken from the lysate, of the pulled down fractions? Do the samples represent the same number of cells in the different cell lines?

We have amended the method section (Antibodies and Western blotting) to provide more details on the input and eluate sample preparation and the amount loaded on gels (lines 699-718). The figure legends for Figure EV1B and D, and Figure 1D have also been amended to indicate that an equivalent number of parasites or amount of protein was loaded for each cell line.

4) Figure EV6 - how did the authors generate the sequence alignment that was based on the secondary structures? What program did they use? What do they mean by indicating that there were missing residues from the caspase-7 structure (cleaved loop) in the Arabidopsis AtRAPTOR1? Were they actually missing, or did

they fail to align? Are these residues missing also from the Leishmania protein? Could these residues account for the missing proteolytic activity in the parasite RAPTOR1?

Original Figure EV6 is not of a structural alignment: we assume the reviewer is referring to original EV4 (which has now been moved to Appendix S1).

In this diagram the sequence alignment shown is not based on sequence similarity but instead the structural overlay of the two proteins. This was achieved using PDBfold with SSM which is part of EBI and ALINE for the drawing of the structural elements. Details and references have been added to the Appendix S1 Figure legend and Materials and Methods (lines 803-807) to clarify this.

The residues are only missing from the caspase structure (not the protein) - so they are not visible in the structure. This is assumed to be a result of the residues being cut at the cleavage site making the loop flexible (Q196-K212, exclusive) and hence not amenable to detection in a crystal structure. The residues are present in this position in AtRAPTOR1. There is no crystal structure available for LmRAPTOR so it is not known if the residues are present but there does not appear to be any conservation of the cleavage sites in the sequences. The text in Appendix S1 Figure legend has been modified to help clarify this.

5) Text: Page 10 The authors should rephrase their conclusions on the requirement for the catalytic function of the assumed proteolytic activity of RPTOR. Although no proteolytic activity was shown, the authors concluded that such predicted activity was not required for promastigote proliferation and growth, since replacement of the missing RAPTOR1 with a protein that was mutated in its cys269 residue enabled the phenotypic recovery of the cells. Do the authors have any evidence that the caspase-7 domain is functional in the Leishmania RPTOR1? Can the authors provide a predicted function for the proteolytic activity of the protein? Is there any other evidence for such an activity other than the structural homologies, that may be misleading?

Unfortunately we were unable to purify *Leishmania* RPTOR1 protein to study proteolytic activity. We have amended the manuscript text to remove the suggestion that it has proteolytic activity. We now refer to Cysteine 269 as not being required for promastigote proliferation.

6) The title: The authors write that nutritional depletion leads to "premature differentiation from proliferative promastigotes to non-dividing mammalian-infective metacyclic forms. These parasites cannot develop into proliferative amastigotes in the mammalian host, or respond to nutrients to differentiate to proliferative retroleptomonads, which are required for their blood-meal induced amplification in sand flies and enhanced mammalian infectivity". In such case the title that declares that TORC1 is an essential regulator of nutrient-dependent differentiation is not fully accurate, since the differentiation process varies from that observed in reality. I would therefore rephrase the title more carefully.

We have amended the title to "TORC1 is an essential regulator of nutrient-controlled proliferation and differentiation in *Leishmania*".

Dear Dr. Myburgh,

Thank you for the submission of your revised manuscript to our editorial offices. I have now received the reports from two of the three referees that I asked to re-evaluate your study, you will find below. Referee #3 was unresponsive to my invitations to assess the revised manuscript, but going through your p-b-p-response I consider his/her points as adequately addressed. As you will see, the other two referees now fully support the publication of the study in EMBO reports. Referee #2 has a remaining concerns, I ask you to address in a final revised manuscript. Please also provide a response to the referee point in a final p-b-p-response.

Moreover, I have these editorial requests I ask you to address:

- Please make sure that the number "n" for how many independent experiments were performed, their nature (biological versus technical replicates), the bars and error bars (e.g. SEM, SD) and the test used to calculate p-values is indicated in the respective figure legends (for main, EV and Appendix figures) of the final revised manuscript. Please also check that all the p-values are explained in the legend, and that these fit to those shown in the figure. Please provide statistical testing where applicable. Please avoid the phrase 'independent experiment', but clearly state if these were biological or technical replicates. Please also indicate (e.g. with n.s.) if testing was performed, but the differences are not significant. In case n=2, please show the data as separate datapoints without error bars and statistics. See also: <http://www.embopress.org/page/journal/14693178/authorguide#statisticalanalysis>

If n<5, please show single datapoints for diagrams. In particular:

Please define the annotated p values ****/**/* in the legend of figure 5b, e as appropriate.

Please indicate the statistical test used for data analysis in the legends of figures 5b, e.

- Please add scale bars of similar style and thickness to the microscopic images (main and EV figures), using clearly visible black or white bars (depending on the background). Please place these in the lower right corner of the images themselves. Presently, the scale bars in Fig. 1D are rather thin and hard to see. Please also remove the size definition for the scale bars from Fig. 5A (just define this in the legend).

- Please remove the info on pre-publication access for referees from the data availability section and add a direct links to the datasets. Please also make sure the data are public latest on the day of publication of the study.

- Table EV1 is a dataset. Please upload the original excel file as dataset, with a legend on the first TAB and name the file Dataset EV1. Finally, please update the callouts ('Dataset EV1') in the main manuscript text file.

- Please move tables EV2 (primer) and EV3 (plasmids) to the Appendix. Please name these Appendix Table S1 and Appendix Table S2, add these to the table of contents, and add a title and a legend. Finally please change their callouts in the main manuscript text file.

- Please remove the legends for the Appendix figures from the table of contents (TOC) of the Appendix. This should only list the items, their titles, and the page. It is sufficient if the legend is provided below the figures.

In addition, I would need from you:

- a short, two-sentence summary of the manuscript (not more than 35 words).

- two to four short (!) bullet points highlighting the key findings of your study (two lines each).

- a schematic summary figure that provides a sketch of the major findings (not a data image) in jpeg or tiff format (with the exact width of 550 pixels and a height of not more than 400 pixels) that can be used as a visual synopsis on our website.

I look forward to seeing a final revised version of your manuscript as soon as possible.

Best,

Referee #1:

The authors have implemented the necessary changes, rendering the manuscript publishable.

Referee #2:

The authors have addressed many of the issues raised by the different reviews with new data as well as changes in the text. Overall, the manuscript is generally improved and the finding will be of significant interest to the molecular parasitology community.

However, one of the new figures needs correction. The authors now localize the RTOR1 protein in live cells by expressing a RPTOR1::mNG fusion protein. Promastigotes stages are co-stained with LysoTracker to identify potential localization of TORC complexes with the lysosome. However, LysoTracker only stains acidocalcisomes in promastigotes (which are the most acidic organelles in this stage) and NOT lysosomes (which are poorly acidified in promastigotes). In the middle set of images, RPTOR1::mNG fluorescence is clearly associated with the lysotracker-positive acidocalcisomes, while in the bottom image it is primarily associated with a tubular structure that is likely the lysosome. The latter would need to be verified by staining with a lysosome-specific marker, such as FM-4-64, as well as Mitotracker (to discount possible localization with the tubular mitochondrion). Verification that RPTOR1 is localized to both the acidocalcisomes and lysosomes substantially adds to the interest in this study but needs to be correctly documented. The quality of these images could also be improved.

EMBOR-2023-56840V3: Response to reviewers

In the revised manuscript we have included new experimental data to address the point raised by the referee. We have also included changes according to editorial requests and the author checklist. These additional/revised data have been incorporated in the Main Figures, Expanded View data and Appendix under the headings below.

Our responses are in blue. Changes in the attached manuscript and supplementary files are also highlighted.

Additional/revised data

Figure 1. Characterization of the RPTOR1 containing complex in *Leishmania*

- **Revised Figure 1D:** Localisation of RPTOR1 using live cell fluorescence microscopy
 - FM-4-64 staining has been included to show localisation of RPTOR1 in lysosomes.
 - Thicker scale bars have been added to all images.

Figure 5. RPTOR1 loss induces metacyclogenesis.

- **Revised Figure 5A** – scale bar definition removed from figure
- **Revised figure 5E** – statistics have been removed for C11 and C12 DMSO (due to 2 data points in each of these groups)

Table EV1 is now Dataset EV1

New Appendix Table S1 and S2

- Previous Tables EV2 and EV3 have been moved to the appendix as Appendix Table S1 and S2.
- Appendix Table of Content has been updated to include tables and remove figure legends.

Point-by-point Responses to reviewers

Referee #2:

However, one of the new figures needs correction. The authors now localize the RTOR1 protein in live cells by expressing a RPTOR1::mNG fusion protein. Promastigotes stages are co-stained with LysoTracker to identify potential localization of TORC complexes with the lysosome. However, LysoTracker only stains acidocalcisomes in promastigotes (which are the most acidic organelles in this stage) and NOT lysosomes (which are poorly acidified in promastigotes). In the middle set of images, RPTOR1::mNG fluorescence is clearly associated with the lysoTracker-positive acidocalcisomes, while in the bottom image it is primarily associated with a tubular structure that is likely the lysosome. The latter would need to be verified by staining with a lysosome-specific marker, such as FM-4-64, as well as

Mitotracker (to discount possible localization with the tubular mitochondrion). Verification that RPTOR1 is localized to both the acidocalcisomes and lysosomes substantially adds to the interest in this study but needs to be correctly documented. The quality of these images could also be improved.

We thank the reviewer for pointing this out. We have now amended Figure 1D and included images of parasites stained with FM 4-64 – this shows that RPTOR1 co-localises with FM 4-64 indicating that the protein localises to the lysosome. The manuscript text (methods in lines 730-748, results in lines 127-136, discussion in lines 438-447 and the figure legend in lines 1237-1240) has been amended to include the new data.

Dr. Elmarie Myburgh
York Biomedical Research Institute, University of York
Hull York Medical School
Wentworth Way
York, I am not in the U.S. or Canada YO10 5DD
United Kingdom

Dear Dr. Myburgh,

I am very pleased to accept your manuscript for publication in the next available issue of EMBO reports. Thank you for your contribution to our journal.

Yours sincerely,
